# LGI3/2–ADAM23 interactions cluster Kv1 channels in myelinated axons to regulate refractory period

Nina Kozar-Gillan[1], Atanaska Velichkova[1], George Kanatouris[1], Yael Eshed-Eisenbach[2], Gavin Steel[1], Martine Jaegle[3], Eerik Aunin[3], Elior Peles[2], Carole Torsney[1,4], and Dies N. Meijer[1,5]

**Along myelinated axons, Shaker-type potassium channels (Kv1) accumulate at high density in the juxtaparanodal region, directly adjacent to the paranodal axon–glia junctions that flank the nodes of Ranvier. However, the mechanisms that control the clustering of Kv1 channels, as well as their function at this site, are still poorly understood. Here we demonstrate that axonal ADAM23 is essential for both the accumulation and stability of juxtaparanodal Kv1 complexes. The function of ADAM23 is critically dependent on its interaction with its extracellular ligands LGI2 and LGI3. Furthermore, we demonstrate that juxtaparanodal Kv1 complexes affect the refractory period, thus enabling high-frequency burst firing of action potentials. Our findings not only reveal a previously unknown molecular pathway that regulates Kv1 channel clustering, but they also demonstrate that the juxtaparanodal Kv1 channels that are concealed below the myelin sheath, play a significant role in modifying axonal physiology.**

## Introduction

Voltage-gated ion channels (VGICs) shape the physiological properties of neurons and neuronal circuits. Their vital role in normal nervous system function not only depends on their intrinsic biophysical properties but also on their surface density and cellular distribution. The high-density accumulation of VGICs in myelinated axons is governed by cellular interactions between the myelinating glial cell and its associated neuron and underlies the rapid, energy-efficient transmission of action potentials (Rasband and Peles, 2021). While voltage-gated sodium channels (Nav1) and high voltage-activated potassium channels (Kv3 and Kv7) are found at high density in the Node of Ranvier (NOR)—the narrow axonal membrane between adjacent myelin sheaths—low voltage-activated Shaker-type potassium channels (Kv1.1 and Kv1.2 in particular) are sequestered under the myelin sheath in the juxtaparanodal (JXP) membrane, separated from the nodal sodium channels by the paranodal axon–glia junction (PNJ). Kv1.1 and Kv1.2 form hetero- and homo-tetrameric K$^+$ channels and are associated with cytoplasmic β-subunits (mainly Kvβ2) that are important for cell surface expression and gating kinetics of the channels (Trimmer, 2015; Gu et al., 2003). It has been suggested that JXP Kv1 channels contribute to stabilization of the nodal membrane potential, prevention of reentrant

currents during development and suppression of backfiring of action potentials at nerve terminals (Zhou et al., 1999; Vabnick et al., 1999; Chiu and Ritchie, 1984). However, the role of these channels under the myelin sheath in normal axonal physiology has remained elusive (Arancibia-Carcamo and Attwell, 2014).

Kv1 channels are mainly expressed in axons. In myelinated axons, they accumulate at the JXP into multi-protein complexes that consist of, in addition to their β-subunits, the cell adhesion molecules Contactin-associated protein2 (CASPR2), Contactin2 (CNTN-2, also called TAG-1), ADAM22 (A Disintegrin And Metalloproteinase22), the membrane-associated guanylate kinases (MAGUKs) PSD93 (DLG2) and PSD95 (DLG4) and the adaptor protein 4.1B (Pinatel and Faivre-Sarrailh, 2020). TAG-1 is present in both the glial inner myelin membrane (adaxonal) and axonal membrane where it interacts with axonal CASPR2, forming a link between the myelin and axonal membranes (Traka et al., 2003; Poliak et al., 2003). Genetic deletion of CASPR2, TAG-1 or protein 4.1B greatly reduces the juxtaparanodal density of Kv1.1/1.2 channels and their associated proteins, whereas deletion of ADAM22 results in the loss of PSD95 and PSD93 from the JXP but leaves all other components intact (Poliak et al., 2003; Traka et al., 2003; Buttermore et al., 2011;

........................................................................................................................................................................................................................
[1]Centre for Discovery Brain Sciences, University of Edinburgh, Edinburgh. UK;   [2]Department of Molecular Cell Biology and Molecular Neurobiology, Weizmann Institute of Science, Rehovot, Israel;   [3]Biomedical Sciences, ErasmusMC, Rotterdam, Netherlands;   [4]Simons Initiative for the Developing Brain, University of Edinburgh, Edinburgh. UK;   [5]Muir Maxwell Epilepsy Centre, University of Edinburgh, Edinburgh, UK.

Correspondence to Dies N. Meijer: dies.meijer@ed.ac.uk.

Cifuentes-Diaz et al., 2011; Einheber et al., 2013; Ogawa et al., 2010). The CASPR2 carboxyl-terminal domain is required for its association with Kv1 complexes at the JXP through a mechanism that involves the actin-binding protein 4.1B, thus linking the complexes to the underlying actin-spectrin network (Horresh et al., 2010, 2008; Zhang et al., 2013). However, the absence of CASPR2, TAG-1 or protein 4.1B does not result in complete loss of JXP Kv1 complexes, suggesting that additional molecules and mechanisms exist that contribute to the accumulation and stabilization of Kv1 complexes (Ogawa et al., 2010; Saifetiarova et al., 2017; Gordon et al., 2014). One such molecule could be the ADAM23 receptor, which we and others have previously described to be present at the JXP (Dhaunchak et al., 2010; Kegel et al., 2014).

ADAM proteins belong to the larger family of zinc metalloproteinases that have diverse functions in development and tissue homeostasis (Seegar and Blacklow, 2019; Hsia et al., 2019; Novak, 2004; Edwards et al., 2008). ADAM23, together with ADAM22 and ADAM11, form a subfamily or clade within the larger family of ADAM proteins and lack metalloproteinase activity (Long et al., 2012). Mutations in these genes are associated with epilepsy, ataxia, spatial memory deficits, and myelination defects (van der Knoop et al., 2022; Muona et al., 2016; Takahashi et al., 2006; Ozkaynak et al., 2010; Sagane et al., 2005). The ADAM11/22/23 proteins have a highly homologous and modular extracellular domain through which they interact with integrins or soluble LGI protein ligands (Kegel et al., 2013). For example, ADAM23 is expressed on antigen-presenting dendritic cells and interacts with the αvβ3 integrin on CD4[+] T-cells to stimulate proliferation and effectiveness of immune response (Elizondo et al., 2016). Within the developing mouse brain, ADAM23 is thought to be involved in axonal arborization through its interaction with LGI1 (Owuor et al., 2009).

All three ADAM proteins have been found associated with Kv1 potassium channel complexes and ADAM11 has been shown to be important for Kv1.1/1.2 clustering at the cerebellar Basket cell terminals that form the pinceau ephapse around the axon initial segment (AIS) of Purkinje neurons (Kole et al., 2015). ADAM23 was found associated with Kv1 channels in presynaptic compartments of excitatory synapses in the hippocampus and it was suggested that ADAM23-LGI1 links to postsynaptic ADAM22, bridging the synaptic cleft and stabilizing AMPA receptor complexes in the postsynaptic membrane (Fukata et al., 2010; Yamagata et al., 2018; Fukata et al., 2006). More recently, a direct role for ADAM23 in modulating Kv1.1/Kv1.4 gated potassium currents was invoked (Lancaster et al., 2019).

We have previously shown that ADAM23 is expressed in Schwann cells and neurons and that ADAM23 accumulates to high density at the JXP, leading us to hypothesize that it is involved in JXP Kv1 complex accumulation and/or stability, possibly through an LGI-dependent mechanism.

Here, we demonstrated that JXP Kv1 channel cluster formation and stability critically depend on axonal ADAM23 and its interactions with its extracellular ligands LGI3 and LGI2. Furthermore, we showed that juxtaparanodal Kv1 channels contribute to the refractory period, thus securing high-frequency firing.

## Results

### ADAM23 expression in myelinated nerves overlaps with Kv1 expression

We and others have previously reported that ADAM23 is highly expressed in the juxtaparanodal domain of myelinated axons in the peripheral and central nerves of young and adult mice (Kegel et al., 2014; Zoupi et al., 2018). The defining characteristic of the JXP is its high density of Kv1 channel complexes. To examine whether ADAM23 and Kv1.1 channels overlap at the JXP, axons were immunolabeled with ADAM23, Kv1.1, and CASPR antibodies. ADAM23 completely overlaps with Kv1.1 at the JXP (Fig. 1 A). Line scans of fluorescence intensity at multiple NORs confirm that Kv1.1 and ADAM23 consistently overlap at the JXP. A representative line scan of a NOR is shown in Fig. 1 A. Kv1 channel complexes are not only found at the JXP but also flank the thin mesaxonal line, which represents the axon–glia junction of the inner mesaxonal tongue (Poliak and Peles, 2003). We examined whether ADAM23 is also expressed at this location and observed ADAM23 immune reactivity directly flanking the inner mesaxonal line, as visualized with CASPR-specific antibodies (Fig. 1 B). Thus, ADAM23 expression overlaps with Kv1 channels at the JXP and mesaxonal line, in line with the suggestion that ADAM23 is a component of the Kv1 channel complexes in myelinated axons.

### ADAM23 is required for Kv1 channel complex accumulation at the JXP

It was previously shown that ADAM22, a molecule closely related to ADAM23, is a component of Kv1 channel complexes and is required for the juxtaparanodal accumulation of the MAGUKS PSD93/PSD95 but is dispensable for Kv1/CASPR2/TAG-1 clustering at the JXP (Ogawa et al., 2010).

To explore whether ADAM23 is an auxiliary or essential component of juxtaparanodal Kv1 protein complexes we examined the expression of Kv1.1, Kv1.2, CASPR2, and ADAM23 in the central and peripheral nervous systems of wildtype and *Adam23*-null (*Adam23^{ΔI/ΔI}* or *A23KO*) mice at 10 d of age (P10, Fig. 1, C and D), the most advanced age before a human endpoint is reached. At P10, virtually all myelin-competent axons in the PNS are myelinated, while in the spinal cord myelination is well advanced. The paranodes of myelinated axons were identified by staining with CASPR-specific antibodies. The node itself was examined using betaIV-spectrin-specific antibodies. As is evident from the representative images in Fig. 1, C and D, there is substantial accumulation of Kv1.1, Kv1.2, CASPR2, and ADAM23 in the JXP, and betaIV-Spectrin in the nodes, of myelinated axons in both the PNS and CNS at this stage of postnatal development. However, in the absence of ADAM23, none of the Kv1 channel complex proteins appear in the JXP.

Thus, it appears that ADAM23 is the major determinant of Kv1 channel complex accumulation at the JXP. We further explored the role of ADAM23 in myelinated axons of the PNS.

### ADAM23 function depends on its interaction with LGI ligands

Next, we asked whether ADAM23 function in targeting or stabilizing the Kv1 channel complexes to the JXP is dependent on its interaction with LGI ligands. ADAM23, like ADAM11 and

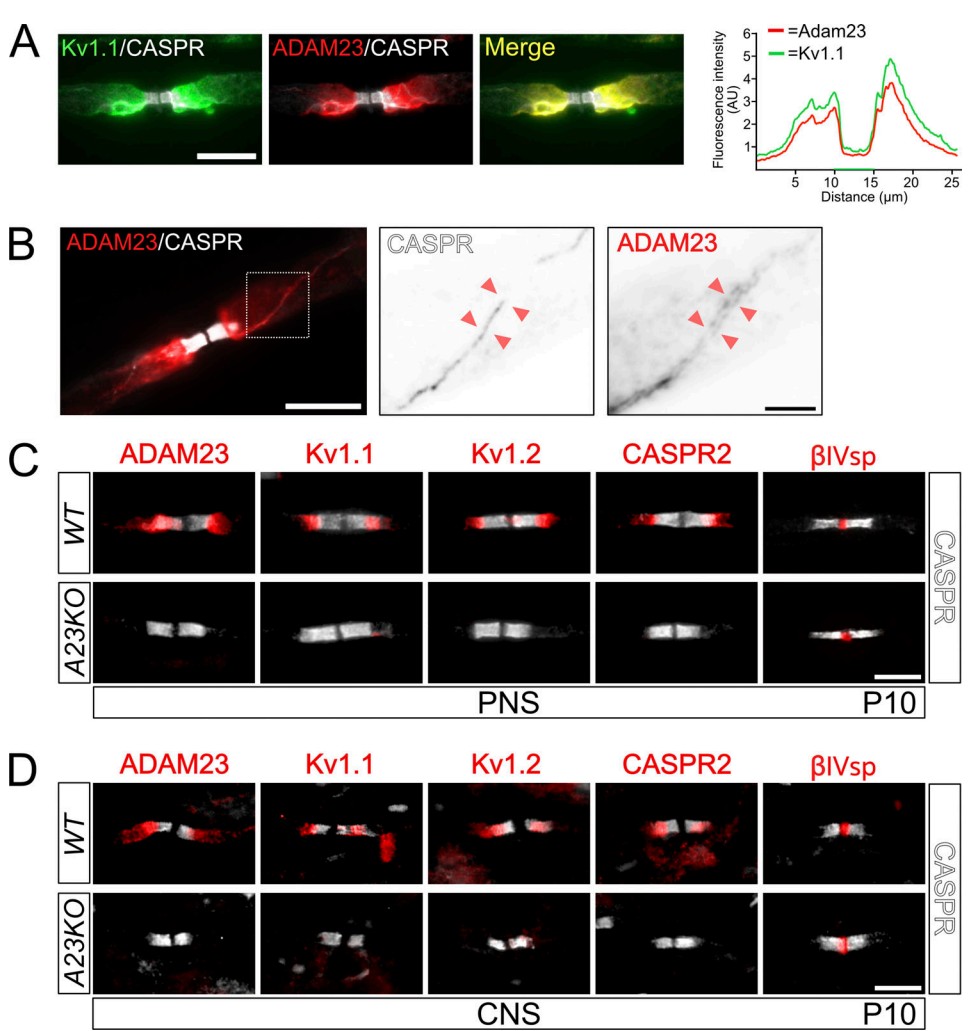

Figure 1. **ADAM23 is required for Kv1 complexes to accumulate at the JXP of myelinated axons in both PNS and CNS. (A)** Left: Immunohistochemistry (IHC) on adult *WT* mouse sciatic nerve axon immunolabeled with antibodies against CASPR (gray), Kv1.1 (green), and ADAM23 (red). Right: Line-scan quantification of fluorescence distribution in ADAM23 and Kv1 staining showing a close overlap in the JXP expression patterns of the two proteins. Scale bar = 10 µm. **(B)** Sciatic nerve IHC on adult mouse teased sciatic nerve axon immunolabeled with CASPR (gray) and ADAM23 (red). Scale bar = 10 µm. Middle and right panels are magnified inverted grayscale images of CASPR and ADAM23 expression in the mesaxonal line region (stippled box in first panel) as indicated by the red arrowheads. Scale bar = 2 µm. **(C and D)** IHC on P10 *WT* (top) and *Adam23^(Δ1/Δ1)* (*A23KO*, bottom) mouse teased sciatic nerve (C) and longitudinal 12 µm spinal cord sections from the cervical region (D). All images contain immunolabeling for CASPR (gray) and either ADAM23, Kv1.1, Kv1.2, CASPR2, or βIV Spectrin (red). Scale bar = 10 µm (applies to all images in C and D).

ADAM22, binds all four members of the LGI protein family (Fig. 2 A (Ozkaynak et al., 2010; Seppälä et al., 2011; Sagane et al., 2008), and all four genes are widely expressed in the nervous system (Bermingham et al., 2006; Herranz-Pérez et al., 2010). We, therefore, first examined which of the four LGI proteins accumulate with Kv1/ADAM23 complexes in the JXP of myelinated axons of the PNS (Fig. 2 B) using antibodies specific for the individual LGI proteins. LGI3 and LGI2 were found to the major LGI proteins present at the JXP. Neither LGI1 nor LGI4 could be detected at the JXP with the available antibodies (Fig. 2 B).

To establish the role of LGI3 and LGI2 in JXP Kv1 complex formation, we generated *Lgi3* and *Lgi2* mutant mice in which the first exon of the *Lgi3* or *Lgi2* gene had been removed (Fig. S1; Marafi et al., 2022). As the first exon encodes the initiation codon and the signal peptide, this strategy resulted in null alleles for both genes. *Lgi3^(Δ1/Δ1)* and *Lgi2^(Δ1/Δ1)* mice are born at normal

Mendelian ratios, are fertile, and do not show any obvious behavioral abnormalities. However, *Lgi2^(Δ1/Δ1):Lgi3^(Δ1/Δ1)* double knock-out animals develop tremors toward the end of the second week of postnatal life and die before weaning. Therefore, mice were analyzed at 12 d of age, before a humane endpoint was reached.

First, we examined whether the deletion of *Lgi2*, *Lgi3*, or both *Lgi2* and *Lgi3* affects myelination of peripheral axons per se. As was the case in the *Adam23^(Δ1/Δ1)* mice (Kegel et al., 2014), myelination of peripheral axons was not affected by deletion of *Lgi2*, *Lgi3*, or both *Lgi2:Lgi3*. The density of myelinated axons in cross-sections as well as myelin thickness appeared normal, and surface area-based g-ratios are all normal for all genotypes analyzed (Fig. 2 C).

Next, Kv1 channel localization at the JXP was examined in *Lgi2*, *Lgi3*, or both *Lgi2* and *Lgi3* deleted mice (Fig. 2 D). Loss of

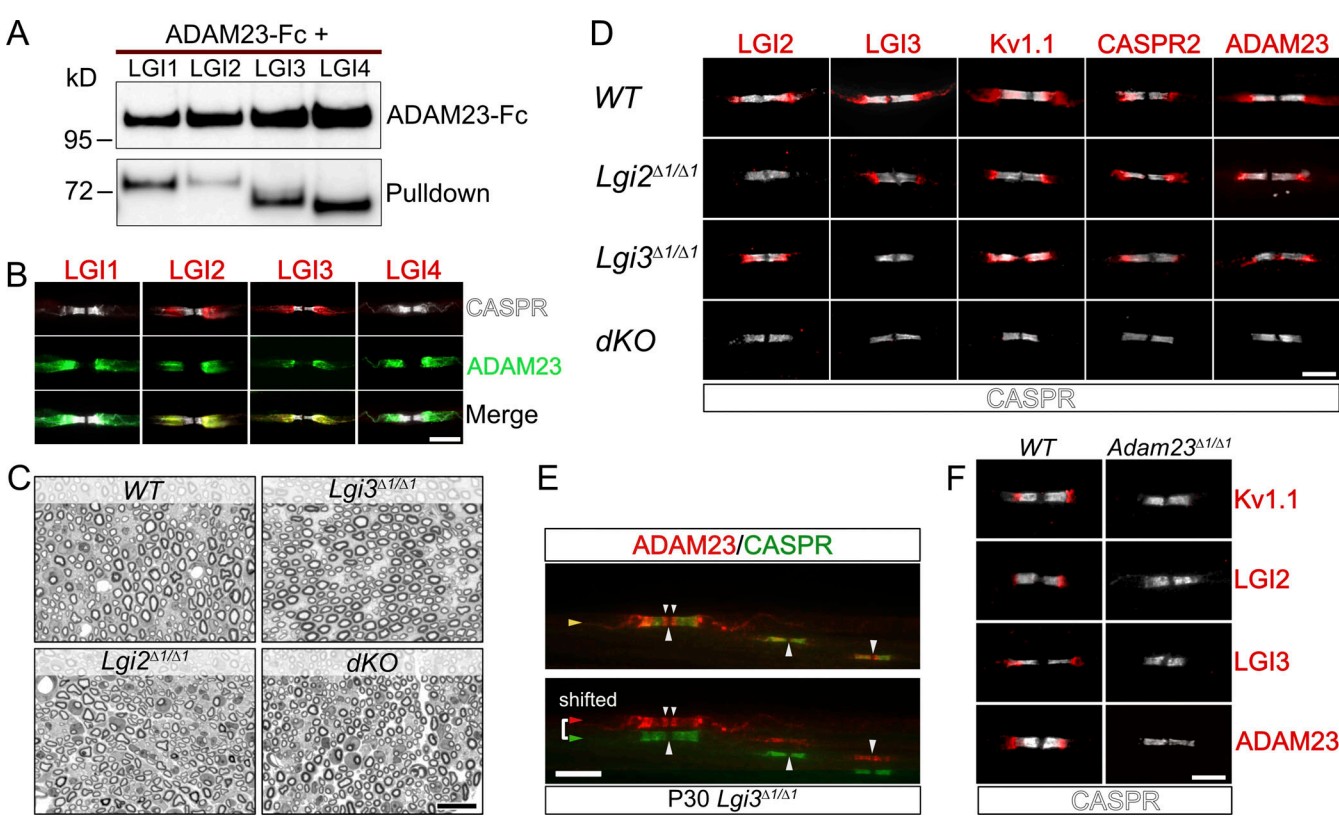

Figure 2. **LGI2 and LGI3 are ligands for ADAM23 at the JXP and play a role in localization of Kv1 channels in myelinated axons. (A)** Western blot (WB) on tissue culture supernatants following co-transfection of HEK293T cells with ADAM23-Fc and one of the four LGI proteins. ADAM23-Fc (top) was precipitated from media with ProtA beads, co-precipitating interacting LGI proteins (bottom, pulldown). **(B)** IHC showing which of the LGI proteins are present at the JXP along with ADAM23. Adult *WT* teased sciatic nerve immunolabeled with CASPR (gray) LGI1, 2, 3, or 4 (red) and ADAM23 (green). Scale bar = 10 μm. **(C)** Semi-thin images of paraphenylenediamine-stained cross sections of sciatic nerves from *WT*, *Lgi3^{Δ1/Δ1}*, *Lgi2^{Δ1/Δ1}*, and double *Lgi2^{Δ1/Δ1}* and *Lgi3^{Δ1/Δ1}* knock-out (*dKO*) adult mice. Scale bar = 10 μm. **(D)** IHC on teased sciatic nerve fibers from P12 mice. *WT* tissue staining shown as a positive control (top) followed by images from mice lacking the expression of *Lgi2* (*Lgi2^{Δ1/Δ1}*), *Lgi3* (*Lgi3^{Δ1/Δ1}*), or both *Lgi2* and *Lgi3* (double knock-out - *dKO*). All images include the paranodal CASPR staining (gray) and one of the following: LGI2, LGI3, Kv1.1, CASPR2, and ADAM23 (red). Scale bar = 5 μm. **(E)** IHC on teased sciatic nerve from an adult (P30) *Lgi3^{Δ1/Δ1}* mouse, stained with antibodies against CASPR (green) and ADAM23 (red). Top image is a merge of both channels whereas bottom image represents split, horizontally shifted signals to indicate mislocalization of ADAM23 and its invasion of the paranodal region in the absence of LGI3 in an adult mouse. Scale bar = 10 μm. **(F)** Examination of LGI2 and LGI3 expression at the JXP in *WT* (left) and *Adam23^{Δ1/Δ1}* (right) P10 mice through IHC on teased sciatic nerve. All images show CASPR (gray) staining and Kv1.1, LGI2, LGI3, or ADAM23 (red). Scale bar = 5 μm. Source data are available for this figure: SourceData F2.

LGI2 does not affect Kv1 channel complex localization at the JXP. However, as we have previously shown, deletion of *Lgi3* profoundly affects Kv1 channel complex localization. In the myelinated axons of *Lgi3^{Δ1/Δ1}* animals Kv1 channels, CASPR2 and ADAM23 are aberrantly localized to the region directly abutting the paranodal domains and infringing on the paranodal domain (Marafi et al., 2022). This mis-localization of Kv1 complexes is a consistent feature in *Lgi3^{Δ1/Δ1}* nerves and does not resolve with age (Fig. 2 E). In stark contrast to the normal juxtaparanodal Kv1 distribution in *Lgi2^{Δ1/Δ1}* animals and mislocalized Kv1 distribution in *Lgi3^{Δ1/Δ1}* animals, deletion of both *Lgi2* and *Lgi3* completely abolishes the accumulation of Kv1 channel complexes at the JXP (Fig. 2 D). These observations suggest that LGI2 function is partially redundant to LGI3.

Although it is evident that ADAM23 does serve as a receptor for LGI2 and LGI3, it is possible that additional, non-ADAM, LGI-receptors such as the NOGO receptor NgR1, exist in the nascent JXP that could initiate accumulation of ADAM23/Kv1 through binding of LGI2 and LGI3 (Thomas et al., 2010). We, therefore, examined LGI2 and LGI3 accumulation at the JXP in P10 nerves of wild-type and *Adam23^{Δ1/Δ1}* animals (Fig. 2 F). No Kv1.1, LGI2, LGI3, or ADAM23 immuno-reactivity (IR) was detected in *Adam23^{Δ1/Δ1}* nerves, whereas all four antigens were readily detected in wild-type controls.

Thus, ADAM23-mediated clustering of JXP Kv1 channel complexes, including CASPR2, requires binding of its ligands, in particular LGI3.

### Neuronal cell-autonomous function of ADAM23 and its ligands

We previously described that ADAM23 is not only expressed in neurons but also in Schwann cells (Ozkaynak et al., 2010). It is, however, not possible to determine with immunofluorescence microscopy whether ADAM23 is present in both the myelin and axonal membrane or only in the Schwann cell membrane at the JXP. If ADAM23 is present in both membranes it is possible that, by analogy with the proposed ADAM22-LGI1/LGI1-ADAM23 interaction that bridges the post- and presynaptic membranes of excitatory neurons in the hippocampus (Fukata et al., 2010;

Yamagata et al., 2018), ADAM23-LGI3/LGI3-ADAM23 bridges the inner myelin membrane and axonal membrane. This hypothesis was tested by deleting *Adam23* specifically in Schwann cells or neurons.

Schwann cell-specific deletion of *Adam23* was achieved using the *Dhh-Cre* driver line, which recombines the LoxP-flanked exon1 of the *Adam23* gene in Schwann cell precursors starting around embryonic day 12 (Kegel et al., 2014; Jaegle et al., 2003). Neuron-specific deletion of *Adam23* was achieved through use of the *Pv-Cre* driver that recombines in parvalbumin-positive proprioceptive neurons and mechanoreceptors (Hippenmeyer et al., 2005; de Nooij et al., 2013). These neurons project large caliber axons (Aα and Aβ afferents) to muscles spindles and Golgi tendon organs and mechanoreceptors in the skin, are heavily myelinated and can be easily identified within sensory roots based on their large diameter. *Pv-Cre* is not active in sensory neurons that project thinner myelinated axons (Aδ afferents).

We first examined the consequence of Schwann cell-specific deletion of *Adam23* on JXP Kv1 channel complexes in sciatic nerve of adult *DhhCre:Adam23$^{LoxP/LoxP}$* (=*Adam23$^{ScKO/ScKO}$*) mice. Kv1 channel complex proteins, including ADAM23 and its ligand LGI3, were found to be normally associated with the JXP membrane in myelinated axons in the sciatic nerve of *Adam23$^{ScKO/ScKO}$* animals (Fig. 3 A). Thus, the Schwann cell expressed ADAM23 is not required for JXP Kv1 complex assembly or stability, and the observed ADAM23 immuno-reactivity must, by inference, represent axonal ADAM23.

In contrast to Schwann cell-specific deletion, neuronal deletion of *Adam23* (*Adam23$^{PvKO/PvKO}$*) resulted in a failure to cluster Kv1 complexes at the JXP. As *Adam23$^{PvKO/PvKO}$* animals develop a gait problem after 6–8 wk these animals were analyzed at 8 wk of age (humane endpoint). None of the Kv1 channel complex proteins were found at the JXP membrane of axons that have lost ADAM23 (solid line boxed nodes in Fig. 3 B), whereas myelinated axons that did express ADAM23 had normal Kv1 channel complexes at the JXP membrane (stipple line boxed nodes in Fig. 3 B). Importantly, no ADAM23 immuno-reactivity was detected in *Pv-Cre* recombined axons, indicating that even if ADAM23 is normally present in the Schwann cell membrane overlying the JXP, its expression there is dependent on neuronal ADAM23.

Additionally, the complete absence of JXP Kv1 channel complexes in myelinated Aαβ axons of 8-wk-old *Adam23$^{PvKO/PvKO}$* animals suggest that no "back up" system exists and that the developmental clustering of these complexes, including CASPR2, is critically dependent on ADAM23 function. Thus, clustering of Kv1 channels requires ADAM23 to be expressed in the axonal membrane.

## ADAM23 is required for Kv1 channel complex stability at the JXP

The mechanisms through which Kv1 channel complexes—once they appear at the JXP—are maintained and stabilized are poorly understood. It has been suggested that direct cis- and trans-interactions between CASPR2 and TAG1 are important to stabilize Kv1 channel complexes at the JXP and that the complexes are linked to the axonal actin/spectrin cytoskeleton through interaction between protein 4.1B and the FERM domain in the cytoplasmic tail of CASPR2 (Traka et al., 2003; Horresh et al., 2008; Poliak et al., 2003).

Having found that clustering of Kv1 channels at the JXP critically depends on ADAM23 and its ligands LGI2 and LGI3, we next asked whether ADAM23 contributes to the stability and maintenance of these complexes. To address this question, *Adam23$^{LoxP/LoxP}$* mice were crossed with *AvCreERT2* mice to obtain two cohorts of defined genotype: *Adam23$^{LoxP/LoxP}$:AvCreERT2* and *Adam23$^{LoxP/LoxP}$*. *AvCreERT2* transgenic mice express the Tamoxifen-inducible Cre recombinase under control of the Advillin regulatory sequences in sensory neurons (Lau et al., 2011). Recombination was initiated in mice at 6 wk of age (Fig. 4 A) and expression of ADAM23 and Kv1 channel complexes in lumbar sensory roots examined at 10 and 29 wk after Tamoxifen administration. Almost two and a half months (10 wk) after initiation of recombination, ADAM23 and Kv1 channels are still found at the JXP, demonstrating the extremely long half-life of the ADAM23 protein in these complexes. However, between week 10 and 29, ADAM23 disappeared from the JXP along with Kv1.1, Kv1.2, and LGI3. Importantly, CASPR2 also disappeared from the JXP within that time frame, demonstrating that its interaction with TAG1 and the actin/spectrin network through protein 4.1B is not sufficient to maintain its position at the JXP in the absence of Adam23. We conclude that Kv1 channel complex stability and maintenance at the JXP critically depend on the continued presence of ADAM23 in the complex.

## Is ADAM23 function dependent on CASPR2?

The lack of detectable levels of Kv1 channel complexes at the JXP in the absence of ADAM23 could result from several different mechanisms involving CASPR2. Given that the earlier literature suggests that CASPR2 is not only involved in stabilizing Kv1 channel complexes at the JXP but might also be required for the initial assembly of these complexes (Traka et al., 2003; Poliak et al., 2003), it is possible that ADAM23 affects the expression of CASPR2 or its accumulation in the axonal compartment, and thus Kv1 channel complex localization at the JXP. A similar mechanism has been described for the Contactin1 (CNTN1)-CASPR interaction which appears crucial for the stable expression of CASPR in the axon (Boyle et al., 2001).

If indeed ADAM23 plays a similar role in directing CASPR2 to the axon, one would predict that the latter, and potentially all other components of the Kv1 complex, do not accumulate in the axon in the absence of ADAM23. To address this question, expression of CASPR2, Kv1.1, Kv1.2, PSD95, and TAG1 was examined in nerves of wild-type and *Adam23$^{ΔI/ΔI}$* animals, by Western blotting and included Neurofilament M chain (NFM) and Myelin Protein Zero (MPZ) as controls (Fig. 5 A). CASPR2 expression levels in sciatic nerve of *Adam23$^{ΔI/ΔI}$* animals do not differ from those of nerves from wild-type animals (Fig. 5 B), demonstrating that ADAM23 does not affect axonal targeting of CASPR2. Additionally, nerve expression levels of Kv1.1, Kv1.2, PSD95, and TAG1 are equally unaffected by the absence of ADAM23 (Fig. 5 B). Therefore, ADAM23 function in Kv1 channel complex clustering at the JXP does not involve axonal targeting of either

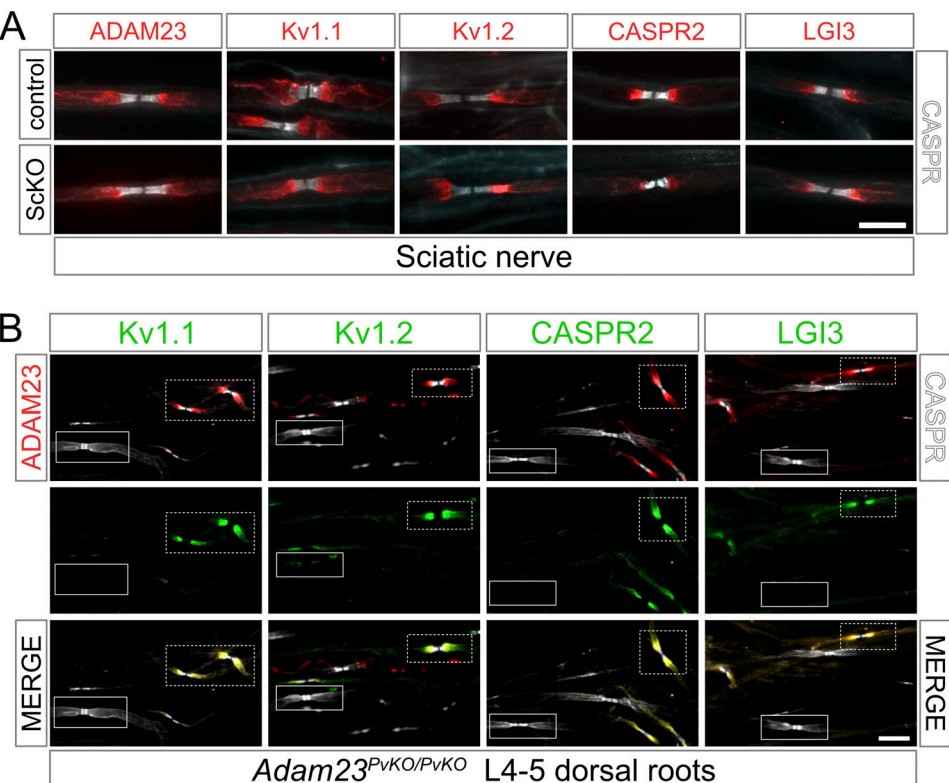

Figure 3. **The function of ADAM23 in Kv1 complex organization depends on the expression of Adam23 in the axonal membrane and not the Schwann cell. (A)** IHC on adult mouse teased sciatic nerve; *Adam23^LoxP/LoxP* (control, left) and *Adam23^ScKO/ScKO* (right) in which *Adam23* is recombined in Schwann cells. Images show merged double staining with CASPR (gray) and ADAM23, Kv1.1, Kv1.2, CASPR2, or LGI3 (red). Scale bar = 10 µm. **(B)** IHC on teased sensory root fibers from lumbar L4-L5 region of the spine of an adult *Adam23^PvKO/PvKO* mouse, where *Adam23* is only recombined in the Parvalbumin (Pv)-positive neurons. Triple immunolabeling with ADAM23 (red) and CASPR (gray; left column), as well as Kv1.1, Kv1.2, CASPR2, and LGI3 (green; middle column). Merged channels are shown in right column. Solid boxes mark large-caliber proprioceptive neurons lacking ADAM23 and the remaining juxtaparanodal proteins. Dashed boxes show the non-recombined axons of the same nerve, which are ADAM23-positive and display normal Kv1 complex expression at the JXP. Scale bar = 25 µm.

CASPR2 or any other established Kv1 channel complex components.

To further define the genetic interaction between ADAM23 and CASPR2 in driving JXP Kv1 complex assembly, we addressed the question of whether ADAM23 is expressed at the JXP through a CASPR2-dependent mechanism. Although the JXP complex expression of Kv1 channels in *Caspr2* knock-out myelinated axons has been examined extensively, these studies mostly involved nerves derived from young adult and older animals (Poliak et al., 2003; Horresh et al., 2010; Saifetiarova et al., 2017). We therefore revisited these expression studies, now using nerves from postnatal day 12 *Caspr2* null (*Caspr2KO*) and wild-type animals. Expression of ADAM23, Kv1.1, Kv1.2, CASPR2, and LGI3 in *wild-type* and *Caspr2 null* nerves was examined (Fig. 6), and it was found that ADAM23 is normally expressed at the JXP in the absence of CASPR2. Additionally, Kv1.1 and Kv1.2 and LGI3 are also normally present at the JXP. These observations, and those presented in Fig. 1, therefore suggest that the initial accumulation of Kv1 channel complexes at the juxtaparanode is dependent on ADAM23 and not CASPR2. Hence, CASPR2 appears not to contribute to initial cluster formation at all, but might instead, together with ADAM23 (see Fig. 4), contribute to the long-term stability of the JXP Kv1 complexes (Saifetiarova et al., 2017; Gordon et al., 2014).

## ADAM23 interactions in JXP Kv1 complex assembly

How then does ADAM23 affect JXP clustering of Kv1 channels? One possibility is that ADAM23 interacts directly with Kv1 channels and, through its extracellular domain, interacts with other members of the JXP Kv1 complexes to pull them into the complex. Interactions between CASPR2 and TAG1 have been documented before (Traka et al., 2003; Horresh et al., 2008; Poliak et al., 2003), but it remains unclear how those interactions affect Kv1 complex formation at the JXP. In this context, it is of interest that it has been suggested that ADAM23 interacts directly with CASPR2 (Hivert et al., 2019). To systematically evaluate potential interactions between the extracellular domain of ADAM23 and other JXP members, a soluble form of ADAM23 (ADAM23γ; Sun et al., 2004) was expressed together with soluble forms (as human IgG1 Fc fusion proteins) of CASPR2, ADAM23, and ADAM22 (Fig. 7 A) and included Neurofascin155 (NF155) as a negative control. NF155 is expressed in the paranodal loops of myelinating glia and interacts with an axonal complex of CASPR and CNTN1. Affinity purification of recombinant proteins from tissue culture media of co-transfected HEK293T cells, followed by Western blotting, revealed that the extracellular domain of ADAM23 directly interacts with ADAM23 itself and ADAM22, but not with CASPR2 (Fig. 7 B). As expected, no

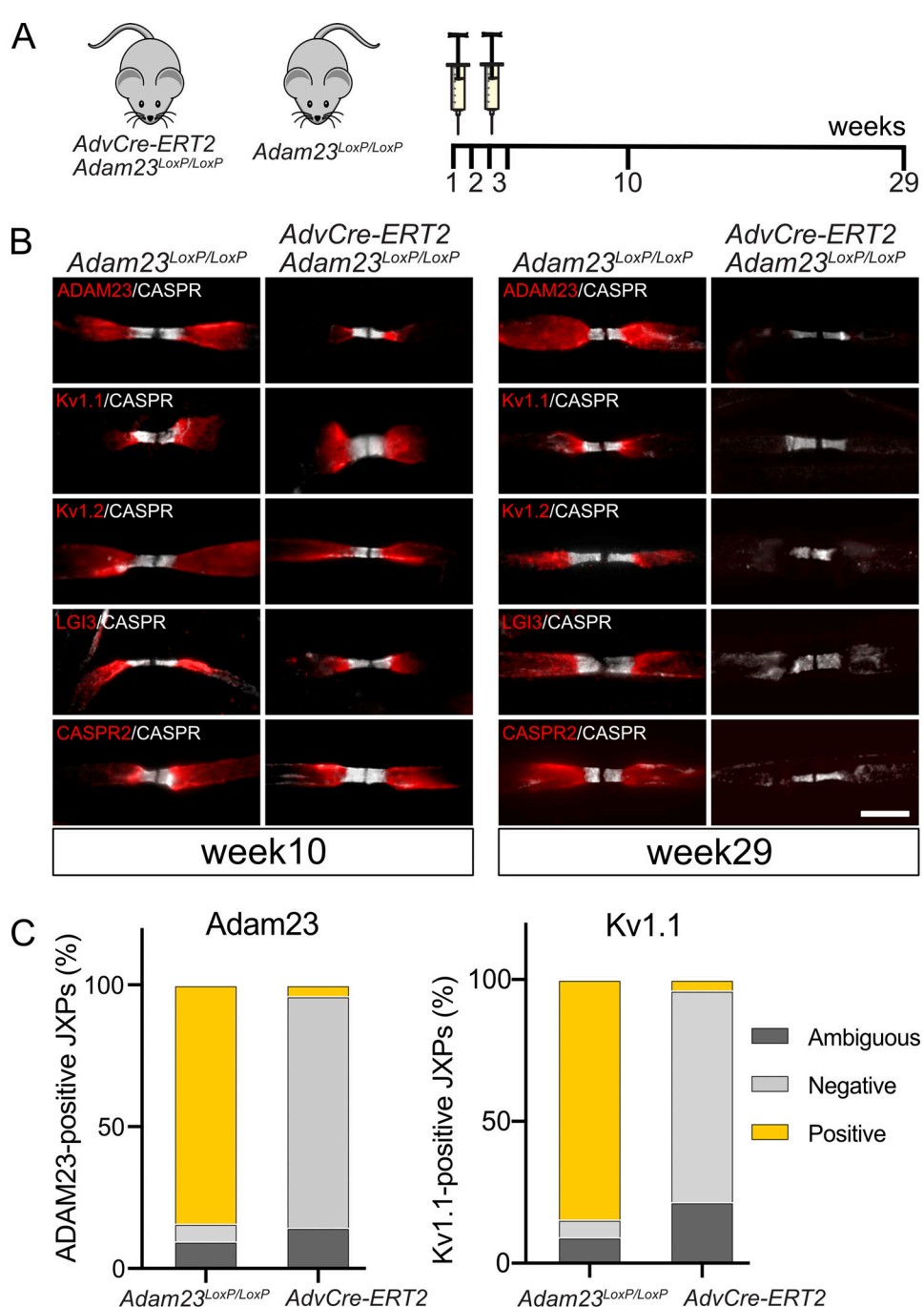

Figure 4. **Loss of ADAM23 from the Kv1 complexes leads to dissociation of the channels from the JXP. (A)** Schematic of tamoxifen administration and tissue collection. Tamoxifen was given to 6-wk-old mice by gavage for five consecutive days, followed by a 1-wk break and administration for another 5 d. Tissue was collected 10 and 29 wks after the first dose of Tamoxifen had been administered. **(B)** IHC on teased sensory nerve axons of adult *Adam23^LoxP/LoxP* and *AdvCre- ERT2:Adam23^LoxP/LoxP* mice after 10 wk (left) and 29 wks (right) from the initial administration of Tamoxifen. All images show paranodal CASPR (gray) and the juxtaparanodal ADAM23, Kv1.1, Kv1.2, LGI3, and CASPR2 (red). Scale bar = 10 μm. **(C)** Quantification of JXPs positive or negative for ADAM23 and Kv1.1 staining 29 wk after Tamoxifen administration. Inconclusive JXPs were denoted as ambiguous. At no instance did we observe Kv1 positive nodes that were not also positive for ADAM23. The paranodal CASPR staining was used as a reference point for counting JXPs. A total of four mice (two per genotype) were analyzed at each time point and an average of 45 nodes were counted per mouse. SD is shown above the relevant bars that represent mean values.

interaction with NF155 was detected. The fact that ADAM23-ADAM23 and ADAM23-ADAM22 interactions can be readily detected in the culture medium suggests these interactions are relatively strong.

To further determine whether the interactions ADAM23 engages in are affected by the presence of the ADAM23 ligand, we repeated these experiments, but now in the presence of co-expressed LGI3. Results shown in Fig. 7 C demonstrate

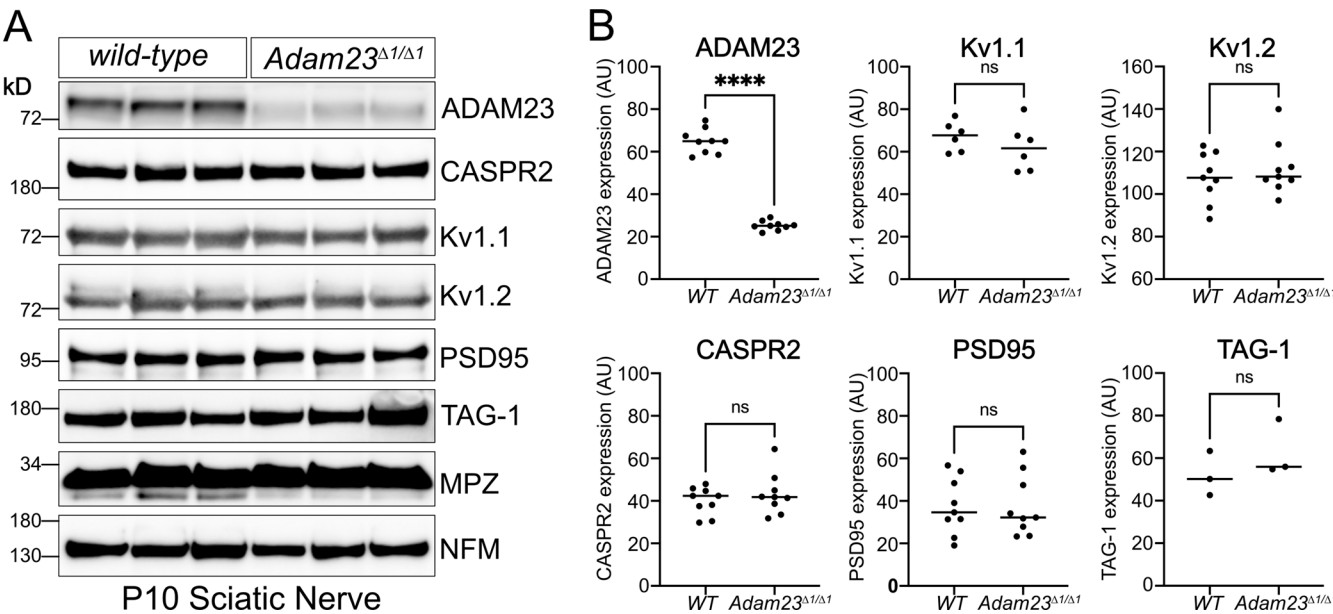

Figure 5. **Axonal expression levels of Kv1 complex proteins in the absence of ADAM23. (A)** Representative Western blot results from homogenized sciatic nerves of P10 *WT* (*Adam23⁺/⁺*) and *Adam23^{Δ1/Δ1}* mice. Each lane represents a pair of sciatic nerves from one mouse. All experiments were performed in triplicate (three biological replicates per genotype per blot). Staining includes ADAM23, CASPR2, Kv1.1, Kv1.2, PSD95, TAG-1, P0 (MPZ), and Neurofilament (NFM). **(B)** Band density for each protein was normalized against P0 and graphed depending on genotype. Each dot represents one mouse, i.e., one pair of sciatic nerves. ADAM23 $n = 9$, Kv1.1 $n = 6$, Kv1.2 $n = 9$, CASPR2 $n = 9$, PSD95 $n = 9$ and TAG-1 $n = 3$. Unpaired $t$ test was carried out to determine statistical significance of the results. ADAM23, $F_{(8, 8)} = 7.0$, ****$P < 0.0001$; Kv1.1, $F_{(5, 5)} = 2.6$, $P = 0.37$; Kv1.2, $F_{(8, 8)} = 1.2$, $P = 0.47$; CASPR2, $F_{(8, 8)} = 2.3$, $P = 0.41$; PSD95, $F_{(8, 8)} = 1.2$, $P = 0.98$; TAG-1, $F_{(8, 8)} = 1.6$, $P = 0.33$. Source data are available for this figure: SourceData F5.

that ADAM23-ADAM23 or ADAM23-ADAM22 interactions appear increased, whereas LGI3 does not appear to mediate an interaction between ADAM23 and CASPR2. The potential enhancement of ADAM23-ADAM23 interactions by LGI3 is of mechanistic importance. To quantify the degree of this enhancement by LGI3, we performed ADAM23-Fc complex precipitations, in triplicate, from media derived from HEK cells co-transfected with ADAM23-Fc and ADAM23γ, with or without LGI3 (Fig. 7 D). Quantitative immunoblotting suggests that LGI3 enhances ADAM23-ADAM23 interactions two- to threefold (Fig. 7 E). It is possible that some of the enhanced complex formation we observe results from increased secretion of transfected proteins in the presence of LGI3. Alternatively, the potential dimerization of LGI3 could result in ADAM23-LGI3 dimer of dimer formation similar to what has been proposed for LGI1 (Yamagata et al., 2018; Yamagata and Fukai, 2020) and thus further enhance complex formation.

### What role do the JXP Kv1 channel complexes play in the physiology of the myelinated axon?

The role of the myelin-covered voltage-gated potassium channels has remained controversial. Chiu and Ritchie (1984) suggested that, following nodal depolarization, internodal Kv channels generate a potassium current that facilitates nodal repolarization. Others have posited that the high resistivity of the multiple myelin lamella and the paranodal pathway effectively insulate these channels (Arancibia-Carcamo and Attwell, 2014). To test whether the juxtaparanodal Kv1 channels indeed contribute to nodal membrane properties, we examined the

excitability, amplitude, conduction velocity, and refractory period of Aαβ sensory fibers in acutely dissected dorsal L4-L5 roots of wild-type and *Adam23^{PvKO/PvKO}* animals (Fig. 8 A). These nerve roots display characteristic Aαβ, Aδ, and C fiber responses (Fig. 8 B). The sensory Aαβ fibers in *Adam23^{PvKO/PvKO}* sensory roots lack the high density of juxtaparanodal Kv1 channel complexes (Fig. 3). The sensory Aαβ-fiber electrical threshold, amplitude, and conduction velocity did not differ between genotypes (Fig. 8, C–E) and were similar to previously published values (Daniele and MacDermott, 2009; Bardoni et al., 2019). However, the relative refractory period is affected and prolonged in the absence of JXP Kv1 channel complexes (Fig. 8, F i and iv). Application of 4-AP, an inhibitor of Kv1.1, Kv1.2, and Kv1.4 channels, to wild-type sensory roots resulted in a similar prolonged refractory period (Fig. 8, ii and K). However, in *Adam23^{PvKO/PvKO}* sensory roots, application of 4-AP does not further prolong the refractory period in these nerves (Fig. 8, G and H, I). Together these results firmly establish that the myelin-covered JXP Kv1 channels do contribute to normal axon physiology by regulating the relative refractory period.

## Discussion

Within both the CNS and PNS, hetero-tetrameric Shaker-type potassium channels (Kv1 or KCNA channels) are predominantly found in the axon initial segment, the axon itself, and at nerve terminals where they act as regulators of excitability and neurotransmitter release (Trimmer, 2015). The most extensively studied members are Kv1.1 and Kv1.2, and mutations or

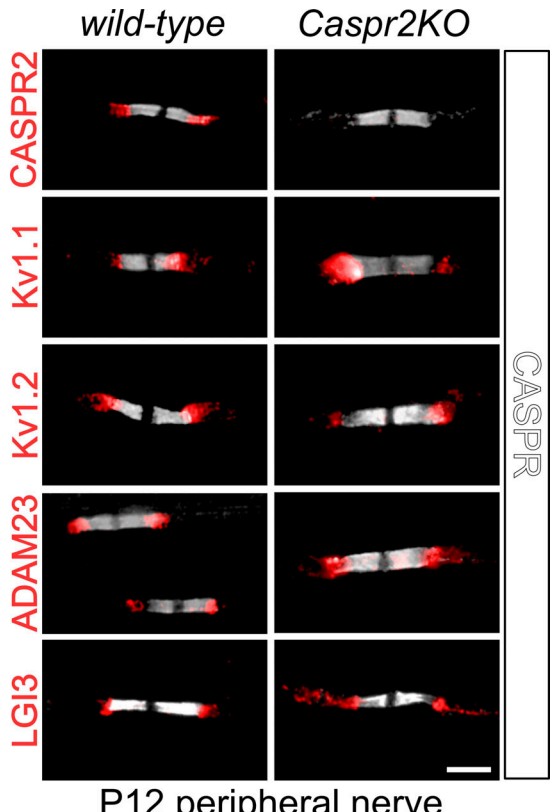

**Figure 6. Organization of JXP Kv1 complexes in postnatal day 12 sciatic nerve axons in wild-type and Caspr2 mutant animals.** IHC on teased sciatic nerve from P12 WT and *Caspr2KO* mice. All images show immune-labeling with CASPR antibodies (gray) and one of the following antibodies; CASPR2 Kv1.1, Kv1.2, ADAM, or LGI3 (red). Scale bar = 10 µm.

autoimmune-mediated dysfunction of these channels result in a diverse range of neurological diseases including episodic ataxia, epilepsy, neuromyotonia, and myokymia (Paulhus et al., 2020; Ovsepian et al., 2016; Irani and Vincent, 2016). These different pathologies reflect Kv1 channel functional diversity which is determined by their biophysical characteristics, density, and exact location in the neuronal membrane. In contrast to un-myelinated axons, where Kv1 channel complexes are evenly distributed in the axonal membrane, in myelinated axons, these channels are found at very high density in the JXP membrane, sequestered under the myelin sheath (Wang et al., 1993). This raises two intriguing questions: What mechanisms drive their accumulation and maintenance at the juxtaparanode, and what function does their concentration in this domain have in normal nerve physiology?

**ADAM23 is essential for the juxtaparanodal accumulation of Kv1 channels**
Here we have demonstrated that ADAM23 is crucially important for formation and stability of JXP Kv1 channel complexes. In the absence of ADAM23, Kv1 channel complexes do not accumulate at the JXP, yet normal levels of Kv1.1/Kv1.2 and their associated proteins PSD95, CASPR2, and TAG-1 are present in the nerves of ADAM23 mutant animals, ruling out a mechanism of action that

involves axonal targeting and co-transport of Kv1 channels and CASPR2 by ADAM23 (Fig. 1 and Fig. 5). It is important to note that even in 6–8-wk-old *Adam23$^{PvKO/PvKO}$* animals (the oldest age that could be examined), no JXP Kv1 channel complexes are detectable, demonstrating that CASPR2 and TAG-1 interactions are not capable of driving the accumulation of the channels in the absence of ADAM23, neither in development nor in adult-hood (Fig. 3). Moreover, deletion of ADAM23 in mature nerves results in a complete disappearance of the Kv1 channels, including CASPR2 and the ADAM23 ligand LGI3, from the juxta-paranode, demonstrating that CASPR2 (in interaction with TAG-1) is not sufficient to maintain Kv1 channels in the absence of ADAM23 (Fig. 4).

A large body of work on NOR formation, and JXP/Kv1 for-mation over the last two decades has led to a model in which CASPR2/TAG-1 interactions and linkage to the underlying cy-toskeleton through CASPR2 cytoplasmic domain interactions with 4.1B was responsible for the accumulation and mainte-nance of the Kv1 channels at the JXP (Poliak et al., 2001; Traka et al., 2003; Horresh et al., 2010; Ogawa et al., 2008; Pinatel et al., 2017; Pinatel and Faivre-Sarrailh, 2020; Poliak and Peles, 2003). Later studies (see also Fig. 6) that demonstrated that a large proportion of developing myelinated axons in Caspr2$^{-/-}$ animals do have detectable levels of Kv1 channels in their nas-cent juxtaparanodes, required modification of this model to one involving alternative mechanisms for clustering Kv1 channels (Ogawa et al., 2010; Saifetiarova et al., 2017). Our results now demonstrate that ADAM23 plays a crucially important role in these mechanisms and that CASPR2 function depends wholly on ADAM23. Interestingly, two studies have shown that in the absence of both CASPR and CASPR2, Kv1 channel complexes are still formed but are now found along the internode (Gordon et al., 2014; Saifetiarova et al., 2017). A similar ectopic inter-nodal location of Kv1 channel complexes is found in aging per-oxisome deficient mice that exhibit altered membrane composition around the node/paranode (Kleinecke et al., 2017). Taken together, these observations suggest that JXP Kv1 clus-tering is primarily determined by ADAM23 and that ADAM23-dependent recruitment of CASPR2 into the complex is required for stable linkage with the underlying cytoskeleton. How CASPR2 is drawn into the complex remains unknown as we could not detect any interaction between the extracellular do-mains of ADAM23 and CASPR2 (Fig. 7).

**Role of LGI ligands**
We demonstrate here that LGI3 and LGI2 are the ligands asso-ciated with ADAM23 in myelinated axons. LGI3 proved to be the major ligand with LGI2 being partially redundant to LGI3 (Fig. 2). Interestingly, deletion of *Lgi3* results in a diminished and mis-localized Kv1 channel expression, with Kv1 channels in-fringing on the paranodal domain (Fig. 2, D and E; Marafi et al., 2022), whereas the simultaneous deletion of LGI3 and LGI2 re-sults in the complete absence of JXP Kv1 channels, an exact phenocopy of *Adam23* mutant axons. While it is demonstrated here that complete lack of JXP Kv1 clusters results in lengthening of the refractory period (Fig. 8), the reduction and mis-localization of Kv1 channels observed in *Lgi3* mutant animals is

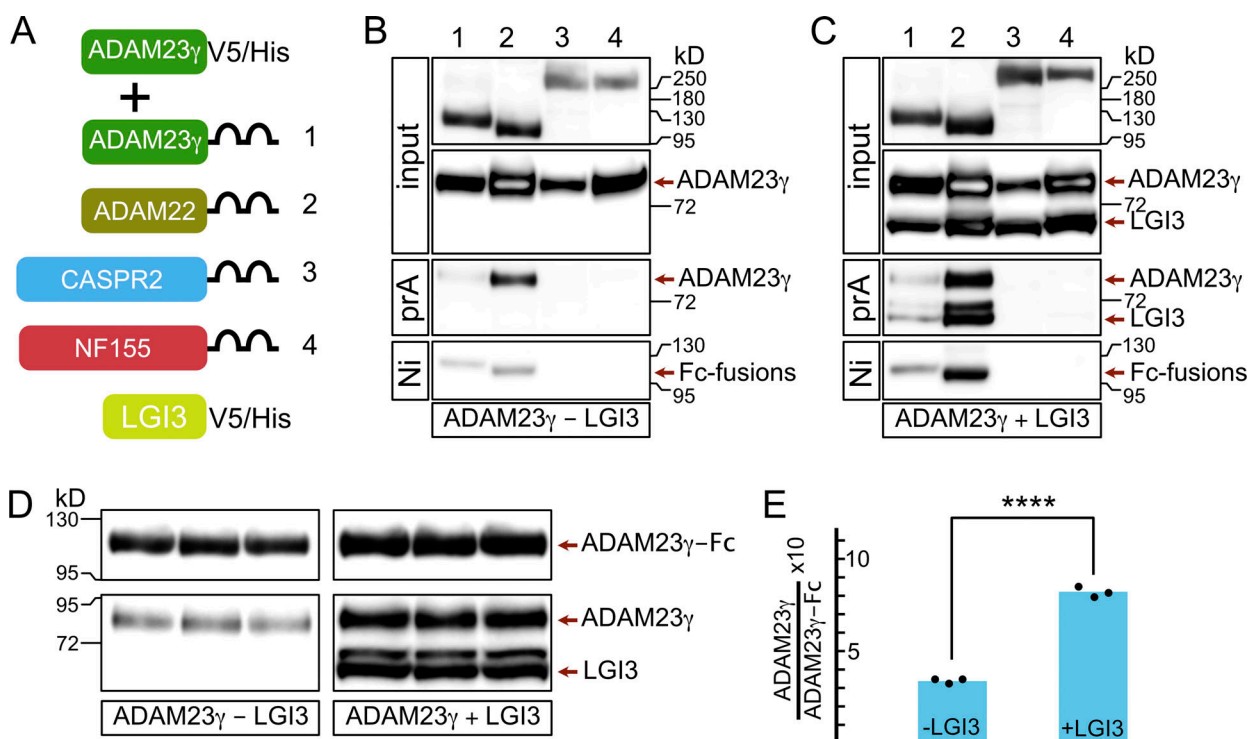

Figure 7. **LGI3 enhances the interactions of ADAM23 with itself and with ADAM22. (A)** Schematic representing simplified structure of DNA plasmids used in HEK293T cell transfections: ADAM23γV5/His and one of the following: ADAM23γFc (1), ADAM22Fc (2), CASPR2Fc (3), or the negative control NFSC155Fc (4). The transfections were done in the presence or absence of LGI3 (LGI3V5/His plasmid). All DNA constructs allowed for production of secreted forms of the relevant protein. **(B and C)** Secreted proteins were precipitated using Protein A (prA) and NiNTA (Ni) beads in LGI3-negative (B) and LGI3-positive (C) transfections and analyzed by Western blot. Top two panels show the input of Fc-fused proteins and ADAM23γHis. The following panel shows which of the Fc-fused proteins co-precipitated, and therefore interacted, with ADAM23γHis. The reciprocal interaction—co-precipitation of Fc-fused proteins with Ni beads—is assessed in the bottom panel. **(D and E)** To allow for quantifiable results, transfection with ADAM23Fc and ADAM23γHis, in presence and absence of LGI3 was repeated in triplicate. ProteinA Sepharose precipitated complexes were analyzed by Western blot (D). Quantification of band densities revealed that interaction between ADAM23Fc and ADAM23γHis is enhanced ~2.5-fold in the presence of LGI3. Unpaired t test was used to assess statistical significance *** = P ≤ 0.001. Source data are available for this figure: SourceData F7.

likely to affect axon physiology in a different way. We have recently described a cohort of patients homozygous for inactivating mutations in LGI3 that display facial myokymia and developmental delay (Marafi et al., 2022). Myokymia may result from instability of the facial nerve since their neurons express high levels of LGI3.

Although much progress has been made in understanding the mechanisms that drive node formation in the PNS and CNS, mechanisms that underpin Kv1 channel accumulation at the JXP have remained less well-defined. Three major mechanisms have been described involving (1) the active clustering of Nav channels at the leading edge of the advancing myelin sheath through the action of cell adhesion molecules, (2) the diffusion barrier function of the paranode, and (3) anchoring of nodal and paranodal proteins to the underlying cytoskeleton (Susuki et al., 2013; Rasband and Peles, 2015; Salzer et al., 2008). Unlike Nav channels, Kv1 channels are not accumulated at heminodes but appear to be concentrated somewhat by the advancing heminodes before they appear in the paranodes and are finally excluded from the node and paranode as myelination is complete (Vabnick et al., 1999; Rasband and Peles, 2015; Hivert et al., 2016). Whether the clearance of Kv1 channels from the nodal and paranodal membranes involves lateral diffusion,

displacement, or active endocytic recycling is unknown. A role for endocytic recycling has been proposed for insertion of nodal proteins in mature nodes and it has been suggested that the paranodal region acts as a sorting hub for these and other proteins (Zhang et al., 2012). A recent study demonstrated that ADAM23 is very stable and rapidly removed from the cell membrane into early and recycling endosomes before being recycled back to the cell surface (Souza et al., 2021). It is thus possible that LGI binding to ADAM23 affects its endocytic recycling and the targeting of ADAM23/Kv1 to the correct axonal compartment and its stable maintenance there.

Our results presented in Fig. 7 suggest that one possible mechanism of ADAM23/Kv1 channel clustering involves the enhanced interaction of ADAM23 molecules by LGI3. Surprisingly, it was found that the ADAM23 extracellular domain can interact with itself and with the extracellular domain of ADAM22, but not with CASPR2. The increase in ADAM23-ADAM23 complexes in the presence of LGI3 could result partly from improved secretion of ADAM23, although we did normalize for ADAM23-Fc expression levels. However, another possibility is suggested by the observation that in crystallography studies, LGI1-ADAM22 complexes can form dimers of dimers through dimerization of LGI1 (Yamagata et al., 2018; Yamagata

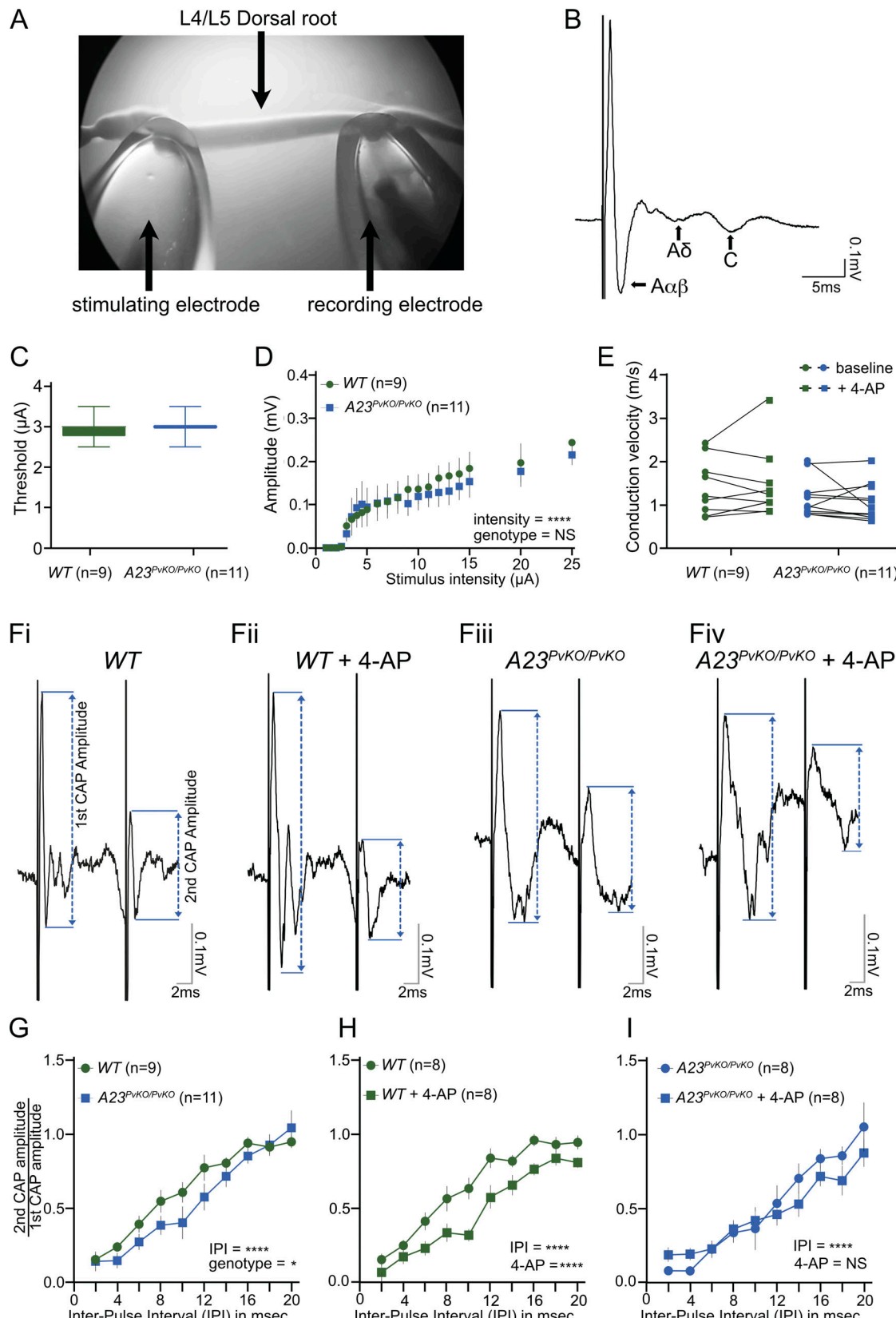

Figure 8. **JXP Kv1 channels do not affect axonal excitability or amplitude but regulate the length of refractory period. (A)** CAP recording set-up, with stimulating and recording glass suction electrodes attached to either end of the L4/L5 dorsal root. **(B)** Representative population CAP recording at 100 μA showing the fast (Aαβ), medium (Aδ), and slow (C) conducting components (arrow), each displaying a characteristic triphasic (positive-negative-positive) response. **(C)** No effect of genotype on Aαβ-fiber activation thresholds (unpaired *t* test, P = 0.44). Sample size: wild-type (WT) dorsal root *n* = 9 (three mice);

*Adam23PvKO/PvKO* dorsal root *n* = 11 (three mice). **(D)** Stimulus-response curve showing increasing Aαβ amplitude with increasing stimulus intensity with no significant effect of genotype on Aαβ amplitude (two-way ANOVA genotype, $F_{(1,342)}$ = 1.02, P = 0.31; intensity, $F_{(18,342)}$ = 6.09, ****P < 0.0001). Sample size: wild-type (WT) dorsal root *n* = 9 (three mice); *Adam23PvKO/PvKO* dorsal root *n* = 11 (three mice). **(E)** Average estimated conduction velocity was not altered by genotype or 4-AP treatment (RM two-way ANOVA, genotype, $F_{(1,18)}$ = 1.85, P = 0.19; 4-AP, $F_{(1,18)}$ = 0.02, P = 0.88). Sample size: wild-type (WT) dorsal root *n* = 9 (three mice); *Adam23PvKO/PvKO* dorsal root *n* = 11 (three mice). **(F)** Representative Aαβ traces at a 10-ms inter-pulse interval (IPI) illustrating the amplitudes of the 1st and 2nd Aαβ CAP in dorsal roots from WT mice (Fi), WT mice + 4-AP(Fii), *Adam23PvKO/PvKO* mice (Fiii) and *Adam23PvKO/PvKO* mice + 4-AP(Fiv). Double-headed arrow illustrates the Aαβ-fiber amplitude. **(G)** Decreasing IPI decreased the 2nd CAP/1st CAP ratio in both WT (*n* = 9 dorsal roots/3 mice) and *Adam23PvKO/PvKO* (*n* = 11 dorsal roots/3 mice) nerves in a genotype-dependent manner (RM two-way ANOVA: genotype, $F_{(1,170)}$ = 6.54, *P = 0.01; IPI, $F_{(9,170)}$ = 32.39, ****P < 0.0001), revealing a significant increased refractory period in Aαβ fibers of *Adam23PvKO/PvKO* mice **(H)** 4-AP decreased the 2nd CAP/1st CAP ratio in WT mice (RM two-way ANOVA: 4-AP, $F_{(1,134)}$ = 45.28, ****P < 0.0001, IPI, $F_{(9,134)}$ = 47.17, ****P < 0.0001). WT (*n* = 8 dorsal roots/3 mice) vs. WT + 4-AP (*n* = 8 dorsal roots/3 mice). **(I)** 4-AP did not alter the 2nd CAP/1st CAP ratio in *Adam23PvKO/PvKO* mice (RM Two-way ANOVA: 4-AP, $F_{(1,127)}$ = 0.77, P = 0.38; IPI, $F_{(9,127)}$ = 18.55, ****P < 0.0001). *Adam23PvKO/PvKO* (*n* = 8 dorsal roots/3 mice) vs. *Adam23PvKO/PvKO* + 4-AP (*n* = 8 dorsal roots/3 mice).

and Fukai, 2020). It is likely that also LGI3 can dimerise while bound to ADAM23, as amino acids involved in LGI1 dimerization are largely identical in LGI3. Thus, one mechanism through which LGI binding could contribute to cell surface clustering of ADAM23/Kv1 complexes is through dimerization.

The observed interaction between ADAM23 and ADAM22 extracellular domains might also explain how ADAM22 is drawn into the JXP ADAM23/Kv1 complexes. And while ADAM22 does not appear to be required for JXP Kv1 clustering in the CNS, it is noteworthy that the ADAM22 carboxyterminus contains a PDZ ligand through which it interacts with MAGUKs such as PSD95 (Lovero et al., 2015). In fact, it was demonstrated that ADAM22 is required for the accumulation of the MAGUKs PSD95 and PSD93 to the JXP (Ogawa et al., 2010). Hence, it is possible that the recruitment of ADAM22 into the JXP Kv1 complex contributes to the stability of the complex through its interaction with MAGUKs (Horresh et al., 2008).

Although we have not fully addressed the cellular origin of LGI3 and LGI2, it is noteworthy that both genes are highly expressed in primary sensory and spinal motor neurons of the PNS and at very low levels in Schwann cells (Bermingham et al., 2006; Ozkaynak et al., 2010). Indeed, deletion of *Lgi3* in Schwann cells only (Fig. S2) does not affect Kv1 channel complex localization. We therefore suggest that the main cellular origin of the LGI ligands in the PNS is the neuron itself and that LGI3/LGI2 act in an autocrine fashion. This stands in contrast with the autocrine and paracrine action of LGI1 in regulating AMPA and NMDA receptor ratios at excitatory synapses in the hippocampus (Lovero et al., 2015) and the paracrine action of LGI4 (through binding of neuronal ADAM22) in promoting myelination in the PNS (Ozkaynak et al., 2010). However, it is important to note that *Lgi3* is highly expressed in oligodendrocytes (Cahoy et al., 2008 and Allen Brain Atlas), suggesting that in the CNS, oligodendrocyte-derived LGI3 could contribute to JXP Kv1 channel complex clustering in a paracrine fashion. Whether this is indeed the case is subject of future studies.

### Juxtaparanodal Kv1 channels contribute to the refractory period of the axon

Whether JXP Kv1 experience activating voltage changes during nodal depolarization critically depends on the existence and resistivity of a conductive path between the periaxonal space (the space between the axonal membrane and glial membrane) and ground. The existence of such a conductive pathway was first suggested by Barrett and Barrett (Barrett and Barrett, 1982), who described a depolarizing afterpotential in myelinated axons from frogs and lizards. They attributed this slowly decaying afterpotential to a passive capacitive current and speculated that this current flows from the internodal perixonal space through the paranodal junctions to extracellular space. Further work suggested that the internodal potassium channels increase their opening after nodal depolarizations and thus contribute to the electrical behavior of the myelinated axon (David et al., 1993; Chiu and Ritchie, 1984; Zhou et al., 1999). These observations and more recent studies indeed strongly suggest that the paranodal junctions do not completely insulate the periaxonal space from the nodal region (Cohen et al., 2019). Indeed, it was demonstrated that compound action potentials were broadened in sciatic nerve of mice homozygous for a Kv1.1 null allele (Smart et al., 1998). However, Kv1.2 channels are still present in the JXP of myelinated axons of Kv1.1 knock out animals, complicating direct comparison with our findings described here (Glasscock et al., 2012). Nevertheless, our results corroborate and extend these findings, as we demonstrate that the absence of high densities of Kv1 channels (both Kv1.1 and Kv1.2) in the JXP of Aαβ proprioceptive axons in our Adam23PvKO/PvKO mice, lengthens the refractory period. Moreover, we demonstrated that a similar lengthening of the refractory period in wild-type Aαβ was achieved by the Kv1 inhibitor 4-AP and that 4-AP does not further lengthen the refractory period in mutant axons, strengthening our conclusion that the sequestered JXP Kv1 channels are activated and contribute to the shape of the action potential. In Adam23 mutant axons, Kv1.1 and Kv1.2 expression levels are not altered, with the Kv1 channels most likely evenly distributed in the internodal membrane. It is thus not just the total number of internodal Kv1 channels, but their high density at the JXP that is required to generate sufficient current to affect action potential shape.

Although we cannot rule out that in the absence of ADAM23 or its ligands LGI3/2, the paranodal junctions are subtly altered to affect the observed electrophysiological properties, we think this unlikely for two reasons. First, disruption of paranodal axon–glia junctions results in reduced nerve conduction velocities (Bhat et al., 2001; Boyle et al., 2001). We do not observe a drop in NCV. Second, application of 4-AP has the same effect on the refractory period as deletion of ADAM23, suggesting that potential alterations in paranodal function do not affect the refractory period.

Different results for 4-AP sensitivity of axons in the PNS versus CNS have been reported (Devaux et al., 2002, 2003). During postnatal development, young rat sciatic nerve axons exhibit re-entrant activity in the presence of 4-AP, but this effect fades with developmental age as Kv1 channels become excluded from the nodal and paranodal membranes and minimal broadening of the AP was observed in mature adult axons (Vabnick et al., 1999; Kocsis et al., 1983). These data suggest that Kv1 channels play an important role in stabilizing axonal membrane potentials during development, but once the axons have adopted a fully mature structure, with Kv1 channels concealed by myelin and separated from the nodal membrane by tight paranodal junctions, 4-AP has very little effect. In contrast, reports on CNS nerves demonstrated variable but significant effects of 4-AP on optic nerve and the ventral funiculus action potentials (Poliak et al., 2003; Devaux et al., 2003). This 4-AP sensitivity of optic nerve has been tentatively attributed to the presence of the Shaw-type Kv channel Kv3.1b in CNS nodes but not in PNS nodes. However, the effect of 4-AP was not altered in *Kv3.1b* mutant optic nerve, leaving open the possibility that hypothetical differences in paranodal structure between CNS nodes and PNS nodes exist that result in differential exposure to voltage changes of the juxtaparanodal Kv1 channels.

Results described here contrast with those obtained from sciatic nerve of *Caspr2*⁻/⁻ and *4.1B*⁻/⁻ animals (Poliak et al., 2003; Cifuentes-Diaz et al., 2011) in which the refractory period was not changed. In these animals JXP Kv1 complexes are greatly reduced or absent from a fraction of NORs (Saifetiarova et al., 2017), yet no broadening of action potential or re-entrant excitation was observed. This difference could possibly be attributed to the complete absence of JXP Kv1 channels in *Adam23*^PvKO/PvKO^ proprioceptive axons versus altered Kv1 expression in the JXP of sciatic nerve axons in *Caspr2*⁻/⁻ and *4.1B*⁻/⁻ animals. However, we cannot rule out that also in the PNS, axons with different motor and sensory modalities exhibit slightly different contributions of JXP Kv1 channels to AP shape. It is of note that Aα and Aβ fibers exhibit very high burst rates (200–800 Hz; Wellnitz et al., 2010; Prochazka et al., 1977) when activated, and it is plausible that the high density of Kv1 channels in the JXP of these fibers facilitate these initial high firing rates. Whether proprioception is affected in Adam23 mutant animals is a question that will be addressed in future experiments.

The potential differential sensitivity of AP shape to altered or absent JXP Kv1 channels in different myelinated axons is further illustrated by a study by Scott and colleagues (Scott et al., 2019). They demonstrated that in Caspr2 null mice, long-range myelinated axons in the corpus callosum, in contrast to sciatic nerve axons (Poliak et al., 2003), exhibit a longer refractory period than wild-type control axons.

In conclusion, we have identified a molecular pathway that regulates JXP Kv1 channel density and position in myelinated axons, thus attenuating basic axonal properties. We demonstrated that LGI ligand-dependent ADAM23 clustering is essential for the targeting and stability of these JXP Kv1 channel complexes. By what mechanism ADAM23 itself is directed to the nascent JXP in the first place is a major unresolved question. We speculate that it involves the preferential segregation of ADAM23/Kv1 complexes in the lipid raft-like axonal membrane that characterizes the paranodal/nodal axolemma environment and stabilization or retention through lateral interactions and LGI ligand binding and dimerization.

## Materials and methods

### Antibodies, plasmids, and chemicals
All antibodies, plasmids, oligonucleotide primer sequences and chemicals, and other resources used in this study are listed in Table S1.

### Mice
Mice used in this project were bred and maintained at the Little France animal unit at the University of Edinburgh. C57BL6/J mice of both sexes were used in all experiments. Animal care and procedures were carried out in accordance with the UK Animals (Scientific Procedures) Act of 1986 (ASPA) and the University of Edinburgh Ethical Review policy. Mice were housed in temperature- and humidity-controlled environment with 12-h light/darkness cycles, cage enrichment, and food and water available ad libitum.

### *Adam23*^Δ1/Δ1^, *Lgi2*^Δ1/Δ1^, *Lgi3*^Δ1/Δ1^, *Lgi2*^Δ1/Δ1^:*3*^Δ1/Δ1^
The *Adam23*^Δ1/Δ1^, *Lgi2*^Δ1/Δ1^, *Lgi3*^Δ1/Δ1^, *Lgi2*^Δ1/Δ1^:*3*^Δ1/Δ1^ genotypes were generated in the Meijer lab using a standardized gene recombination approach, as previously described (Kegel et al., 2014; Jaegle et al., 2003). The method is summarized here using *Lgi2*^Δ1/Δ1^ as an example (see Fig. S1). A LoxP site was inserted upstream of the first exon whilst an Frt-flanked neo cassette with a 3′ LoxP site was inserted downstream, resulting in the *Lgi2*^LoxP^ allele. The *Lgi2*^Δ1^ null allele was then generated through crossing *Lgi2*^LoxP/+^ mice with mice carrying germline Cre recombinase leading to deletion of the first exon and its promoter. The *Lgi2*^Δ1/+^ mice were eventually intercrossed, generating *Lgi2*^Δ1/Δ1^ offspring. Mice of the *Lgi2*^Δ1/Δ1^ and *Lgi3*^Δ1/Δ1^ genotypes developed normally, and their lifespans and fertility were not affected by the mutation. *Adam23*^Δ1/Δ1^ and *Lgi2*^Δ1/Δ1^:*3*^Δ1/Δ1^ mice expressed severe phenotypes, characterized by poor postnatal growth, body tremors seen from second post-natal week, and early lethality (around P12 in the *Adam23*^Δ1/Δ1^ mice and P16 in the *Lgi2*^Δ1/Δ1^:*3*^Δ1/Δ1^).

### *AvCreERT2*:*Adam23*^LoxP/LoxP^
To generate the *AvCreERT2*:*Adam23*^LoxP/LoxP^ mouse line, *AvCreERT2* mice were bred with *Adam23*^LoxP/LoxP^ mice (Lau et al., 2011). Tamoxifen activation of Cre-recombinase expressed in Advillin-positive cells resulted in deletion of exon 1 of *Adam23*. Tamoxifen was administered to 6-wk-old mice by gavage at 1.2 mg per 10 g of body weight, for five consecutive days. This was followed by a 1-wk break, after which administration was repeated for another 5 d. Animals were culled and tissue was collected at appropriate time points after Tamoxifen administration.

### *PvCre*:*Adam23*^LoxP/LoxP^
In this mouse model, the Cre-mediated recombination of *Adam23* was tissue-specific and confined to Parvalbumin (Pv)-positive

cells such as the interneurons in the brain and the large-diameter proprioceptive afferent sensory neurons of the dorsal root ganglia (de Nooij et al., 2015). The parvalbumin promoter of the Cre knockin allele directs Cre recombinase expression in Pv-expressing cells (Hippenmeyer et al., 2005). The allele, originally denoted as $Pvalb^{tm1(cre)Arbr}$ is referred to here as $PvCre$. Mice expressing $PvCre$ were crossed with $Adam23^{LoxP/LoxP}$ mice, leading to Cre-mediated recombination of $Adam23$ in Pv-positive tissue. $PvCre:Adam23^{LoxP/LoxP}$ mice are referred to as $Adam23^{PvKO/PvKO}$.

### $DhhCre:Adam23^{LoxP/LoxP}$
Schwann cell-specific deletion of $Adam23$ was achieved by crossing the $DhhCre$ transgene into the $Adam23^{LoxP/LoxP}$ (Jaegle et al., 2003; Kegel et al., 2014).

### Genotyping
The Polymerase chain reaction (PCR) method was used to determine mouse genotypes using ear or tail biopsies and the primer oligonucleotides listed in Table S1. PCR reactions were performed in 1xPCR buffer (Promega), 5 mM of each primer, 10 mM dNTPs, and nuclease-free water. PCR program conditions were as follows: denaturation (98°C, 1 cycle, 30 s), denaturation (98°C, 35 cycles, 5 s), annealing (various temperatures, 35 cycles, 5 s), extension (72°C, 35 cycles, 10 s), final extension (72°, 1 cycle, 1 min). Annealing temperatures differed between genotypes and were as follows: Adam23 -54–60°C, Lgi2 -58°C, Lgi3 -64°C, PvCre -60°, AvCre -59°.

### Cell culture and transfection
HEK293T cells were cultured in Dulbecco's Modified Eagle Medium (DMEM) supplemented with 10% Fetal Bovine Serum and 1% Penicillin-Streptomycin and maintained in a humidified incubator with 5% $CO_2$ concentration at 37°C, using standard cell culture plastic dishes. Upon reaching ~85% confluence, cells were washed with Dulbecco's phosphate-buffered saline (DPBS) and dissociated in TrypLE Express Enzyme. Once completely detached, cells were resuspended in DMEM and mixed until single cell suspension was achieved. Cells were then distributed at appropriate concentrations into new dishes containing fresh DMEM and placed back in the incubator. All solutions were warmed up in a 37°C water bath before being applied to cells.

In preparation for transfection of HEK293T cells, plastic cell culture dishes were coated with Polyethyleneimine (PEI) in cell culture grade water (20 µg/ml), for ~40 min to 1 h at RT to ensure strong adhesion and even distribution of cells. Once coated, dishes were washed twice with DPBS, and cells in DMEM were seeded at appropriate densities. When cell density reached around 65%, media were refreshed, and cell transfection was performed using PEI MAX transfection reagent maintaining a PEI/DNA ratio of 3:1 in all experiments. Concentration and volume of DNA and reagents were calculated based on the size of the dish. For example, if transfections were carried out in 10 cm dishes, plasmid DNA of total concentration of 7.5 µg was diluted in 0.3 ml of 150 mM NaCl in a sterile Eppendorf tube. In a separate tube, 22.5 µg of PEI-MAX was also diluted in 0.3 ml of 150 mM NaCl and vortexed vigorously. The solutions were then combined, mixed gently, and incubated at RT for 15 min before being added to cells dropwise.

Secreted protein was purified from conditioned medium 24–36 h after transfection. Conditioned media were collected, centrifuged to remove debris and the pH was neutralized by adding 1/10 volume of 0.5M $Na_2PO_4$, pH8. Nickle-charged Ni-NTA agarose beads (Qiagen) were used for purification of proteins containing a polyhistidine tag (6 consecutive histidine residues [6xHis]) while Protein A (ProtA) agarose beads (Repligen) were used for purification of proteins possessing an Fc tag. 60 µl of Ni-NTA or ProtA beads in slurry (30 µl of settled beads) were transferred to 2 ml Eppendorf tubes, washed twice with DPBS, and centrifuged for 5 min at RT, 200 × g. Following aspiration of wash buffer, conditioned media were added to the beads. In NiNTA purification, samples were combined with 1xHisTag buffer diluted from a 10× stock (1× HisTag buffer: 20 mM NaPO4 pH8, 10 mM Imidazole, 150 mM NaCl, 0.1% Triton X-100). Media were incubated with beads for 4 h at RT with constant rotation. Samples were centrifuged for 5 min at 200 × g, and the supernatants were discarded. NiNTA beads were washed with wash buffer (50 mM NaPO4 pH8, 10 mM Imidazole, 300 mM NaCl) while ProtA beads were washed with DPBS followed by a 5-min centrifugation at 200 × g, RT. All liquid was removed, beads were resuspended in 60 µl of 2× LDS sample buffer and incubated for 5 min at 37°C. After a 10 min centrifugation at 5,000 × g, eluted proteins in sample buffer were frozen at −20°C and stored until analysis through Western blotting.

### Immunohistochemistry, image acquisition, and analysis
Where mouse tissue was analyzed through immunohistochemistry (IHC), animals were euthanized by perfusion-fixation. Mice were anesthetized using Sodium Pentobarbital and transcardially perfused with 0.1M NaPO4, pH 7.4, followed by 4% methanol-free paraformaldehyde (PFA) in the same NaPO4 buffer.

Sciatic nerves were collected and fixed in 4% PFA in 0.1M NaPO4, pH 7.4, on ice for 10 min if tissue came from P10 animals and 20 min if harvested from adult mice. Fixation times were further extended by 10 min if the mouse had not been perfusion-fixed. The fixative solution was then replaced with Sörensen's buffer (136.68 mM $Na_2HPO_4$, 33.32 mM $KH_2PO_4$; pH 7.4) in which the nerves were stored until further processing at 4°C.

To tease nerve fibers into single axons, the nerves were transferred onto adhesion microscope slides containing a large drop of Sörensen's buffer and dissected into shorter pieces. The epineural sheath was removed and nerve fragments were teased apart into single axons using acupuncture needles. Teased nerves on slides were air-dried overnight at room temperature and either used immediately or frozen in −20°C until further use.

### Sensory and motor roots
Where ventral and/or dorsal roots were analyzed, spinal columns were harvested following perfusion-fixation. The dissected spinal column was placed in 4% PFA in NaPO4 (pH 7.4) on ice for 1 h for further fixation before being transferred to

Sörensen's buffer. To extract sensory and motor roots, the column was cut in half longitudinally using a surgical scalpel and the motor and sensory nerves of lumbar level 3–5 (L3-5) were dissected using fine forceps and stored separately in Sörensen's buffer at 4°C. Prior to IHC, the sensory and motor fibers were teased into single axons on glass cover slips as described for the sciatic nerve.

### Spinal cord

To obtain spinal cord sections, mice were perfusion-fixed and spines (cervical to lower thoracic region) harvested and fixed as above. Spinal cords were extracted by pulling away each vertebra. For cryoprotection, the extracted spinal cord was placed in sucrose diluted in PBS; 45 min in 7.5% sucrose, followed by 45 min in 15% and an overnight incubation in 30% sucrose. The spinal cord was then positioned in an embedding mold, covered with OCT embedding matrix and frozen on dry ice. Spinal cords in OCT were stored at −80°C before being sectioned on cryostat microtome. Prior to sectioning, the OCT blocks were placed in the cryostat set to −20°C for 1 h to allow for temperature equilibration. Spinal cord sections of 12 μm were collected on adhesion glass microscope slides and air-dried overnight before being processed for IHC or frozen at −20°C.

### Immunohistochemistry

The same basic IHC protocol was used for each of the mouse tissues described above. Tissues were rehydrated with three 5-min washes in PBS and incubated with ice-cold methanol for 10 min at −20°C. Methanol was washed off with another three 5-min washes with PBS. The slides were transferred into a humidified incubation box, tissue was covered with fish skin gelatin (FSG) blocking solution (5% FSG, 0.3% Triton X-100, 1x Sörensen's buffer), and incubated for 1 h at RT. Primary antibodies were prepared in the same blocking buffer and applied at equal concentrations and volumes to experimental and control tissue. Incubation with primary antibodies was done in a humidified box at 4°C overnight. Following incubation, slides were washed six times for 15 min at RT using Tris-Buffered Saline (TBS) containing 0.01% Triton X-100 (10x TBS: 138 mM NaCl, 2.7 mM KCl, 19 mM Tris, $H_2O$, pH set to 7.4). Secondary antibodies were diluted in 5% FSG blocking buffer and incubated on slides for 1 h in a humidified, light-tight box at RT. Slides were then washed six times for 15 min with TBS/0.01% Triton X-100, once with TBS and once with $dH_2O$. Slides were air-dried in a light-tight box at RT and glass cover slips were mounted on top using DAPI-containing Mowiol medium.

### Image acquisition and processing

Samples mounted in Mowiol were imaged using a Zeiss AxioImager-Z1 microscope equipped with a Zeiss AxioCam MR camera using 20× Plan-Apochromat NA = 0.8 air and 63× Plan-Apochromat, NA = 1.4 oil objectives. Secondary antibodies used carried Alexa fluorochrome conjugates Alexa488, Alexa555 and Alexa647 (details on antibodies can be found in Table S1). Mounting medium contained DAPI to allow imaging of nuclei. Images were captured using ZEN2 pro imaging software using the same exposure time for both wild-type and mutant mouse tissue samples. All images were processed and analyzed using Fiji software (ImageJ) and Adobe Photoshop version 24.1.0.

## Western blot analysis
### Sample preparation

P10 $Adam23^{\Delta/\Delta}$ and age-matched WT control mice were culled, and sciatic nerves were harvested into Eppendorf tubes, snap-frozen in liquid nitrogen, and stored at −80°C until further processing. Nerves were homogenized using a Sample Grinding Kit and homogenization buffer (25 mM HEPES, pH 7.4, 150 mM NaCl, 2 mM EDTA, 1 mM $Na_3VO_4$, 2.5 mM $Na_4P_2O_7$, 20 mM NaF, 2.5 mM β-Glycerophosphate, 1% NP40, 1% Tx100, 1% SDS, 1% Deoxycholate, PMSF, PIM). Samples were then centrifuged at 4°C for 10 min at 20,000 × $g$, and supernatants were transferred into fresh tubes. Protein concentration was measured using the Pierce Protein Assay (BCA). Samples of appropriate, equalized protein concentrations and volumes were mixed with LDS Sample Buffer and frozen at −20°C or immediately used in WB.

### Gel electrophoresis and blotting

Samples in LDS buffer were reduced by addition Sample Reducing Agent to 50 mM and heated for 10 min at 70°C. Proteins were separated on a 4–12% Bis-Tris gel using MOPS running buffer for 32 min at 200V in a Bolt electrophoresis tank. Color pre-stained protein standard was loaded on gels alongside samples. Proteins were blotted onto 0.45 μm nitrocellulose membranes (Amersham) for 1 h at 10V, using Bolt transfer buffer combined with methanol and Bolt antioxidant as per producers' guidelines (Invitrogen). After transfer, membranes were washed with $dH_2O$ and air-dried overnight at RT prior to blocking in 10% milk blocking buffer diluted in TBS with 0.05% Tween (TBS-T) for 1 h. This and all following incubations and washes were performed in light-tight LI-COR incubation boxes, at RT with gentle, continuous rocking. Blocked membranes were incubated overnight with appropriate primary antibodies diluted in TBS-T containing 5% blocking buffer. Antibody incubation was followed by 4 times 10 min washes with TBS-T. HRP-coupled secondary antibodies (1:20,000) were diluted in TBS-T with 5% milk block and added to membranes for 1 hr. Membranes were washed 3 times for 10 min with TBS-T and once with TBS. To develop the HRP signal, membranes were incubated for 5 min with West Pico PLUS Chemiluminescent Substrate (Pierce) and imaged using LI-COR OdysseyFc imaging system. Images were analyzed and bands were quantified using Fiji software.

## Electrophysiology
### Isolated dorsal root preparation

Isolated dorsal roots were prepared as described previously (Torsney, 2011; Dickie et al., 2017). Following isoflurane-induced anesthesia, control ($Adam23^{LoxP/LoxP}$; $n = 3$) and knock-out ($PvCre$: $Adam23^{LoxP/LoxP}$; $n = 3$) mice (∼6–8 wk old) were decapitated and their lumbar (L4/L5) dorsal roots were removed in an ice-cold dissection solution. L4/L5 dorsal roots were cut near their entry zone and their ganglia were removed. The roots were briefly recovered for up to 15 min in 32–34 °C oxygenated NMDG recovery solution and then placed in oxygenated NaCl holding solution for

1 h at room temperature prior to recording. For the electrophysiological compound action potential (CAP) recordings, the isolated roots were transferred to a recording chamber of an upright microscope (Zeiss) and perfused with a constant flow (1–2 ml/min) of oxygenated recording solution. The 95% $O_2$/5% $CO_2$-saturated dissection solution contained 3.0 mM KCl, 1.2 mM $NaH_2PO_4$, 26 mM $NaHCO_3$, 15 mM glucose, 251.6 mM sucrose, 7 mM $MgCl_2$, and 0.5 mM $CaCl_2$, pH 7.3–7.4. The recording solution contained 125.8 mM NaCl, 3.0 mM KCl, 1.2 mM $NaH_2PO_4$, 26 mM $NaHCO_3$, 15 mM glucose, 1.3 mM $MgCl_2$, and 2.4 mM $CaCl_2$, pH 7.3–7.4. The NMDG recovery solution comprised 93 mM NMDG. 2.5 mM KCl, 1.2 mM $NaH_2PO_4$, 30 mM $NaHCO_3$ 25 mM glucose, 20 mM HEPES, 5 mM Sodium absorbate, 2 mM Thiourea, 3 mM Sodium pyruvate, 10 mM $MgSO_4$, 0.5 mM $CaCl_2$, pH 7.3–7.4. The holding solution contained 92 mM NaCl, 2.5 mM KCl, 1.2 mM $NaH_2PO_4$, 30 mM $NaHCO_3$, 25 mM glucose, 20 mM HEPES, 5 mM Sodium absorbate, 2 mM Thiourea, 3 mM Sodium pyruvate, 2 mM $MgSO_4$, 2 mM $CaCl_2$, pH 7.3–7.4.

### Compound action potential recording

The ex vivo CAPs were recorded using two glass suction electrodes placed at each end of the dorsal root, one for electrical stimulation and the other for recording. The three main components of the CAP were distinguished based on their activation threshold and conduction velocity as fast (Aαβ), medium (Aδ), and slow (C) conducting components, each displaying a characteristic triphasic (positive–negative–positive) response. To characterize the Aαβ component, dorsal roots were stimulated three times at 0.2 Hz with an ISO-flex stimulus isolator at 0.5–5 μA (in steps of 0.5 μA), 6–14 μA (in steps of 1 μA), and at 15–25 μA (in steps of 5 μA) with a 0.1-ms pulse width.

To measure the Aαβ refractory period (RP), a paired-pulse (0.1-ms wide) stimulation at 25 μA was delivered to the dorsal root with a gradually shortened inter-pulse interval from 20 to 2 ms (in steps of 2 ms). The Aαβ RP is represented as the ratio of the 2nd Aαβ CAP on the 1st Aαβ CAP amplitude (2nd CAP/1st CAP) as a function of the inter-pulse interval. To assess the contribution of Kv1 channels to the RP, following baseline recordings in both WT and KO mice, 500 μM of the Kv1 channel blocker 4-aminopyridine (4-AP) was bath-applied 10 min prior to, and during, a repeated set of RP recordings.

Data were acquired and recorded using an ER-1 differential amplifier and pClamp 10 software. Data were filtered at 10 kHz and sampled at 50 kHz.

### Quantification and statistical analysis

Quantification of band intensities on Western blots was carried out using ImageJ (Fiji) software. Statistical analysis was done in Microsoft Excel (2019) and GraphPad Prism 9. Graphs were generated in GraphPad Prism. Values in quantification graphs represent the mean and SD, unless indicated otherwise. Data distribution was assumed to be normal but was not formally tested. Statistical tests were chosen according to the experimental design and are indicated in the relevant figure legends. Statistical significance with null hypothesis was determined using the following $P$ values: * = $P \leq 0.05$, ** = $P \leq 0.01$, ***$P \leq 0.001$, **** = $P \leq 0.0001$.

Figures were prepared using ImageJ (Fiji) software, Adobe Photoshop 24.1.0, and Affinity Designer graphic software (Serif Europe).

### Online supplemental material

Fig. S1 shows the map of the mouse *Lgi2* gene and the targeting strategy to create the conditional and null allele. Fig. S2 shows juxtaparanodal expression of Kv1 complex components in wild-type and *Adam23*[ScKO/ScKO] myelinated nerves. Table S1 list all antibodies, reagents, primers, and plasmids as well as software and hardware used in this study.

## Acknowledgments

We would like to thank Annelies van den Bogaard, Siska Driegen, Ella Mercer, and Alex Maas for their assistance in mouse genetic procedures and the initial characterization of *Adam23*, *Lgi2*, and *Lgi3* targeted alleles and Peter Brophy and Elaine Dzierzak for comments on the manuscript.

This work was supported by the UK Research and Innovation Biotechnology and Biological Sciences Research Council (BBSRC) grant number BB/N015142/1 and BB/008008/1 to D. Meijer, and grant number BB/M010996/1 (N. Kozar-Gillan and A. Velichkova), and the Dr. Miriam and Sheldon G. Adelson Medical Research Foundation (E. Peles).

Author contributions: Conceptualization: N. Kozar-Gillan, A. Velichkova, and D. Meijer; investigation: N. Kozar-Gillan, A. Velichkova, G. Kanatouris, E. Aunin, M. Jaegle, G. Steel, and D. Meijer; visualization: N. Kozar-Gillan, A. Velichkova, and D. Meijer; formal analysis: N. Kozar-Gillan, A. Velichkova, C. Torsney, and D. Meijer; resources: E. Peles and Y. Eshed-Eisenbch; writing-original draft: N. Kozar-Gillan, A. Velichkova, and D. Meijer; writing-review and editing: N. Kozar-Gillan, A. Velichkova, E. Peles, and D. Meijer; funding acquisition: D. Meijer; supervision: C. Torsney and D. Meijer. All authors read and approved the final manuscript.

Disclosures: The authors declare no competing interests exist.

Submitted: 7 November 2022

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

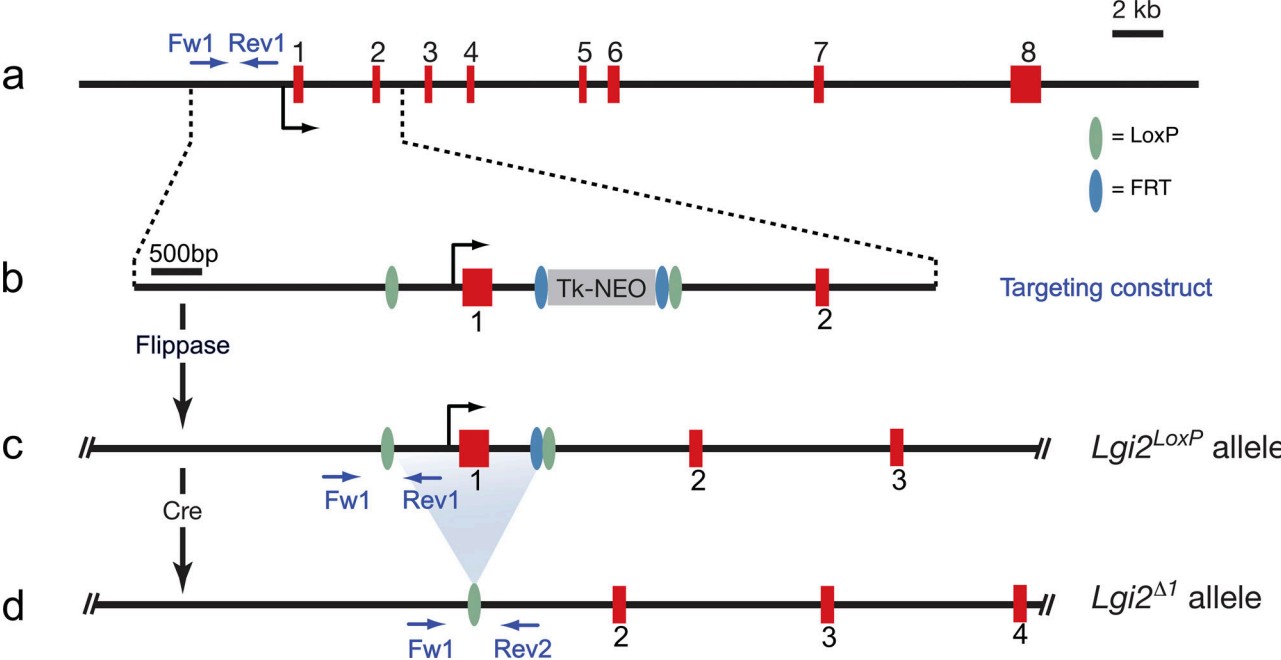

Figure S1. **Targeting strategy to create a *Lgi2* conditional allele and a *Lgi2* null allele. (a)** Structure of the mouse *Lgi2* gene on mouse chromosome 5. The *Lgi2* gene has 8 exons with the first exon encoding the signal peptide, the LLRNT domain and the first half of the first LRR repeat. **(b)** Schematic depiction of the targeting construct which introduces a LoxP upstream of exon 1 and in intron1. The TK promoter-driven Neomycin selection cassette is flanked by FRT sites. **(c)** Mice carrying the targeted *Lgi2* allele were crossed with a transgenic mouse expression the Frt recombinase (Flippase) in the germ line, to generate the *Lgi2LoxP* allele. **(d)** Crossing mice carrying the *Lgi2^LoxP* allele with a germ-line expressing Cre recombinase mice results in the deletion of the first exon of *Lgi2* (*Lgi2^Δ1*).

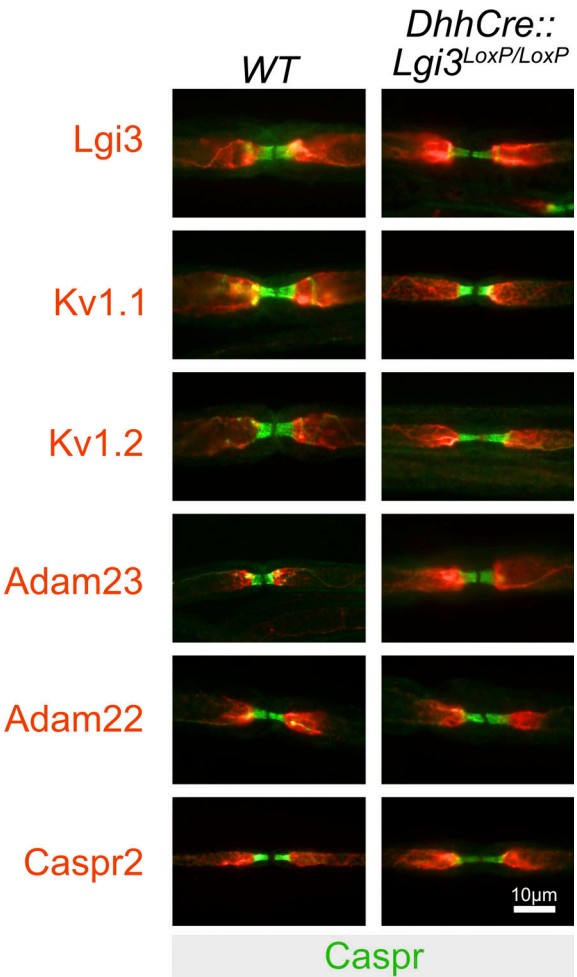

Figure S2. **The node of Ranvier in the PNS of adult wildtype and Schwann cell-specific deleted *Lgi3* mice.** Representative images of nodes of Ranvier in the sciatic nerve of wild-type (WT) and Schwann cell-specific deleted Lgi3 mice (*DhhCre:Lgi3<sup>LoxP/LoxP</sup>*). Nerves were stained with Caspr antibodies in green to visualize the paranodal domains flanking the nodal gap and with either Lgi3, Kv1.1, Kv1.2, Adam23, Adam22, and Caspr2 antibodies to visualize the components of the juxtaparanodal Kv1 complexes. No differences in immunostaining between the two genotypes were observed.

**Provided online is Table S1. Table S1 lists primers and plasmids as well as software and hardware used in this study.**

