## [Peer Review File · The Journal of Cell Biology]

LGI3/2-ADAM23 interactions cluster Kv1 channels in myelinated axons to regulate refractory period

Nina Kozar-Gillan, Atanaska Velichkova, George Kanatouris, Yael Eshed Eisenbach, Gavin Steel, Martine Jaegle, Eerik Aunin, Elijor Peles, Carole Torsney, and Dies Meijer

Corresponding Author(s): Dies Meijer, University of Edinburgh

Review Timeline:

Submission Date:	2022-11-07
Editorial Decision:	2022-12-03
Revision Received:	2022-12-18
Editorial Decision:	2022-12-23
Revision Received:	2023-01-10

Monitoring Editor: Marc Freeman

Scientific Editor: Lucia Morgado-Palacin

Transaction Report:

DOI: <https://doi.org/10.1083/jcb.202211031>

December 3, 2022

Re: JCB manuscript #202211031

Prof. Dies N Meijer
University of Edinburgh
Centre for Discovery Brain Sciences
49 Little France Crescent
Edinburgh EH16 4SB
United Kingdom

Dear Prof. Meijer,

Thank you for submitting your manuscript entitled "LGI3/2-ADAM23 interactions drive clustering of Kv1 channels in myelinated axons to fine-tune axonal properties". The manuscript was assessed by expert reviewers, whose comments are appended to this letter. We invite you to submit a revision if you can address the reviewers' key concerns, as outlined here.

As you will see, the reviewers are enthusiastic about your study. However, they raise some issues that would need to be solved before we can move forward. Rev #1 asks what ADAM23 does at the paranodal junction and how it is restricted to the juxtaparanode. Likewise, rev #3 asks if ADAM23 regulates the relocation of juxtaparanodal components to paranodes and if ADAM23 loss results in paranodal defects.. Rev #2 requests additional clarifications on results and experiments performed and asks whether ADAM23 colocalizes with other Kv1 channels. We find these reviewers' points valid and agree they need to be addressed, with appropriate data where requested. We hope you will be able to address the other reviewers' points too.

GENERAL GUIDELINES:

Text limits: Character count for an Article is < 40,000, not including spaces. Count includes title page, abstract, introduction, results, discussion, and acknowledgments. Count does not include materials and methods, figure legends, references, tables, or supplemental legends.

Figures: Articles may have up to 10 main text figures. Figures must be prepared according to the policies outlined in our Instructions to Authors, under Data Presentation, <https://jcb.rupress.org/site/misc/ifora.xhtml>. All figures in accepted manuscripts will be screened prior to publication.

*****IMPORTANT:** It is JCB policy that if requested, original data images must be made available. Failure to provide original images upon request will result in unavoidable delays in publication. Please ensure that you have access to all original microscopy and blot data images before submitting your revision. ***

Supplemental information: There are strict limits on the allowable amount of supplemental data. Articles may have up to 5 supplemental figures. Up to 10 supplemental videos or flash animations are allowed. A summary of all supplemental material should appear at the end of the Materials and methods section.

Please note that JCB now requires authors to submit Source Data used to generate figures containing gels and Western blots with all revised manuscripts. This Source Data consists of fully uncropped and unprocessed images for each gel/blot displayed in the main and supplemental figures. Since your paper includes cropped gel and/or blot images, please be sure to provide one Source Data file for each figure that contains gels and/or blots along with your revised manuscript files. File names for Source Data figures should be alphanumeric without any spaces or special characters (i.e., SourceDataF#, where F# refers to the associated main figure number or SourceDataFS# for those associated with Supplementary figures). The lanes of the gels/blots should be labeled as they are in the associated figure, the place where cropping was applied should be marked (with a box), and molecular weight/size standards should be labeled wherever possible.

The typical timeframe for revisions is three to four months. While most universities and institutes have reopened labs and

allowed researchers to begin working at nearly pre-pandemic levels, we at JCB realize that the lingering effects of the COVID-19 pandemic may still be impacting some aspects of your work, including the acquisition of equipment and reagents. Therefore, if you anticipate any difficulties in meeting this aforementioned revision time limit, please contact us and we can work with you to find an appropriate time frame for resubmission. Please note that papers are generally considered through only one revision cycle, so any revised manuscript will likely be either accepted or rejected.

Thank you for this interesting contribution to Journal of Cell Biology. You can contact us at the journal office with any questions, cellbio@rockefeller.edu or call (212) 327-8588.

Sincerely,

Marc Freeman
Monitoring Editor
Journal of Cell Biology

Lucia Morgado-Palacin, PhD
Scientific Editor
Journal of Cell Biology

Reviewer #1 (Comments to the Authors (Required)):

Kozar-Gillan et al. use a combination of genetically modified mice, imaging, and biochemistry to further elucidate the enigmatic clustering mechanisms for juxtaparanodal K⁺ channels. These channels have been of much interest because their assembly and clustering is a terrific example of axon-glia interactions regulating the organization of the axonal membrane. In addition, the molecules involved all have human pathogenic variants that converge on epilepsy and other neuropsychiatric diseases. The manuscript clearly shows the key component of the juxtaparanodal complex required for Kv1 channel clustering and maintenance is axonal ADAM23. This is a major advance and forces a reconsideration of the previous mechanisms (previously it was thought that Caspr2 and TAG1 were the key molecules). The authors also clearly demonstrate how LGI2 and LGI3 participate in the complex and can facilitate stronger interactions among ADAM proteins. The experiments are very well done and very convincing, and the paper is very easy to read and follow. I think the paper is highly appropriate for JCB and I'm quite enthusiastic. I have only a few minor comments for the authors to consider:

1. Regarding the refractory period: Smart et al., Neuron 1998 showed that in Kv1.1 knockout mice the refractory period of sciatic nerve is increased. I think the authors should reference this paper as it presages the conclusions from the experiments in ADAM23 KO studies.
2. The authors have not considered whether loss of ADAM23 may also subtly affect the paranodal junction. Loss of LGI3 appears to somehow affect paranodal junction function allowing K⁺ channels and ADAM23 into the paranodal region (Fig. 2D, E). Would a subtle disruption like this alter refractory period? This may also be a point for discussion.
3. One major question that remains unanswered and not addressed here is how ADAM23 is restricted to the Juxtaparanode in the first place. It is clear that clustering of Kv1 channels, Caspr2, etc. all depend on ADAM23, but what gets ADAM23 there in the first place? Do the authors have any suggestions for how this might happen? I realize this is beyond the scope of the paper, but it seems that the authors might speculate or at least comment on how this might happen.
4. IPI is not defined in the manuscript.

Reviewer #2 (Comments to the Authors (Required)):

In this manuscript Kozar-Gillan et al investigate mechanisms by which Kv1 channels are clustered at the juxtaparanodal region and the function they serve there. They demonstrate using genetic mouse models that axonal expression of the metalloproteinase Adam23 is required for initial clustering as well as stability of Kv1 at juxtaparanodes via interaction of Adam23 with LGI2 and LGI3. They also show that conditional knockout of neuronal Adam23 and a consequential lack of juxtaparanodal Kv1 channels results in impaired refractory period, suggesting a function for these juxtaparanodal Kv1 channels. This is a rigorously conducted, well-written study on a topic of interest to the field. I have only minor comments:

- 1) In Figure 1, only colocalization of Kv1.1 with Adam23 was examined. Does Adam23 also colocalize with other Kv1 channels?

- 2) It would be helpful if the authors provide an expanded discussion of how their findings relate to previous studies of Kv1 knockout mice.
- 3) The CASPR2 immunostaining in Figure 3A looks like it might be reduced in the Adam23 ScKO. Is that the case?
- 4) Figure 3B does not show staining in L4-5 sensory roots from control animals, only the (nice) internal control of non Cre-expressing Adelta fibers. It would be best to also be able to compare the PV+ axons from cKO and control animals.
- 5) Could the authors expand on the method by which PV+ axons were identified in Figure 3B? This seems important as there appear to be about 10% of axons without or with ambiguous Adam23 immunostaining JXPs at 16 weeks of age (Figure 4B), so the lack of Adam23 may not be sufficient to identify the axon as PV+. The authors state axon diameter is used to identify PV+ axons, but the exact criteria is not given. Even better, was costaining done to confirm these axons are in fact PV+ axons?
- 6) The CASPR staining appears more diffuse in the Adam23 PvKO PV+ neurons in Figure 3B. Is that the case?
- 7) Some of the text references for Figure 2 appear to be incorrect.

Reviewer #3 (Comments to the Authors (Required)):

The present manuscript focuses on the mechanisms underlying Kv1 channel clustering in myelinated axons and the functional role of this channel regarding axonal electrophysiological properties. This work shows that ADAM23 is required for both accumulation and maintenance of Kv channel complexes at the juxtaparanodes, and that this depends on ADAM23 interaction with its ligands LGI3/2. It further shows that juxtaparanodal Kv channels affect the refractory period, suggesting that they could participate in enabling the high frequency firing of action potentials. This is an interesting study, in particular as there was until now a lack of clear data on juxtaparanodal Kv channel role in axonal conduction properties. The manuscript is very well written and illustrated. I have only few comments and suggestions regarding this work.

1. Kv channel accumulation at paranodes is slightly delayed compared to paranodal and nodal clustering during development. It would have been interesting to show whether ADAM23 follows a similar pattern, by looking at slightly younger ages (P9 and P6?). Also, could Kv1 briefly accumulate in the nodal area prior to be retrieved or to diffuse in ADAM23 KO?
2. Some of the WT images (ADAM23, Figure 1C and Kv1.2, Figure 1D) are suboptimal.
3. Line 189: why look at P12 and not P10 as before? This question applies for other experiments implicating one of these two timepoints along the study. This should be clarified in the text.
4. Line 200: the mislocalization of juxtaparanodal markers in the paranode could suggest a paranodal alteration. This should maybe be discussed. Also, with aging, juxtaparanodal components tend to relocate to paranodes. Could LGI3/AMDAM23 be implicated in this process?
5. Page 14 : the notion of axonal targeting may confuse the readers. The authors should maybe clarify that it does not refer to axonal membrane targeting.
6. Regarding the role of JXP Kv1 complex in the physiology of the axon, can a paranodal defect be excluded in ADAM23PvKO/PvKO animals, as it could participate in electrophysiological alterations ?

Minor comments:

Could the authors discuss the lack of compensation between ADAM11 and ADAM23?

There is a typo line 213.

Rebuttal JCB manuscript #202211031

The authors thank the three reviewers for their thoughtful and helpful comments. All the points are addressed below. Following suggestions, textual changes have been made are highlighted in red in the revised manuscript. Figure 3 is revised in response to a comment of reviewer 2.

Reviewer #1

Kozar-Gillan et al. use a combination of genetically modified mice, imaging, and biochemistry to further elucidate the enigmatic clustering mechanisms for juxtaparanodal K⁺ channels.The manuscript clearly shows the key component of the juxtaparanodal complex required for Kv1 channel clustering and maintenance is axonal ADAM23. This is a major advance and forces a reconsideration of the previous mechanisms.... The experiments are very well done and very convincing, and the paper is very easy to read and follow. I think the paper is highly appropriate for JCB and I'm quite enthusiastic. I have only a few minor comments for the authors to consider:

1. Regarding the refractory period: Smart et al., Neuron 1998 showed that in Kv1.1 knockout mice the refractory period of sciatic nerve is increased. I think the authors should reference this paper as it presages the conclusions from the experiments in ADAM23 KO studies.

A very valid point and we apologise for the omission. We have now added the reference and text in the discussion section. However, it is important to point out that Kv1.1 knock out animals cannot be directly compared with the Adam23KO, as Kv1.2 channels are present at the JXP of myelinated axons in Kv1.1 knock out animals (Glosscock et al., 2012).

2. The authors have not considered whether loss of ADAM23 may also subtly affect the paranodal junction. Loss of LGI3 appears to somehow affect paranodal junction function allowing K⁺ channels and ADAM23 into the paranodal region (Fig. 2D, E). Would a subtle disruption like this alter refractory period? This may also be a point for discussion.

We agree this is a valid point for discussion. Disruption of paranodal junction, by deletion of for example Caspr or Contactin, results in the presence of juxtaparanodal components, including ADAM23 (unpublished data), in the paranodal domain. Such disturbance, in all cases examined, results in strong reduction of nerve conduction velocities (Bhat et al., 2001; Susuki et al., 2007; Boyle et al., 2001) and other less well characterised electrophysiological abnormalities. We note that NCV in sensory roots is not affected by genotype (Figure 8). In addition, examination of the sciatic nerve NCV at P10 in wildtype and Adam23KO animals shows no statistically significant differences (Maria Fjeldstad and Dies Meijer, unpublished observations). These results are compatible with a normal functioning paranodal junction.

As the reviewer notes, in Lgi3 mutant nerves, a fraction of Kv1 channel complexes is found in the paranodal domain (see also Marafi et al., 2022). This could be interpreted as a subtle defect of the paranodal junction in Lgi3KO mice. However, it could equally be indicative of defect in a LGI3/ADAM23-dependent mechanism that normally clears Kv1 channels complexes from the maturing paranode. We have examined the nerve conduction velocity of wildtype and Lgi3KO animals during development and found that it is normal at all stages of postnatal development (Maria Fjeldstad, Nina Kozar-Gillan and Dies Meijer; unpublished observations).

Thus, despite the altered location of Kv1 complexes in Lgi3KO animals, the normal developmental increase in NCV is observed, again, suggesting that paranodal development and function is not affected by the Lgi3 mutation.

3. One major question that remains unanswered and not addressed here is how ADAM23 is restricted to the Juxtaparanode in the first place. It is clear that clustering of Kv1 channels, Caspr2, etc. all depend on ADAM23, but what gets ADAM23 there in the first place? Do the authors have any suggestions for how this might happen? I realize this is beyond the scope of the paper, but it seems that the authors might speculate or at least comment on how this might happen.

This is a very difficult question to answer, and we can only speculate. As the initial accumulation of Kv1 channel complexes appears to follow the formation of the node and maturation of the paranodal junctions, it seems plausible that ADAM23 axonal membrane localisation depends on axolemma/axonal cytoskeleton specialisations imposed by the paranodal/nodal structures. In this respect it is of note that specific gangliosides are enriched at the node and altering the composition of these gangliosides by genetic means results in nodal alterations that can result eventually in the disruption of the JXP and abnormal localisation of Kv1 channel complexes (Kleinecke et al., 2017; Susuki et al., 2007). A similar drifting away of Kv1 channel complex from the JXP is observed in Caspr/Caspr2 double knock outs (Gordon et al., 2014; Saifetiarova et al., 2017) demonstrating that paranodal junction formation and anchoring to the axonal actin/spectrin network is

essential for JXP localisation. And although the presence of ADAM23 in these ectopic Kv1 complexes is not directly demonstrated it is plausible that they are part of these complexes. We thus propose that 1) ADAM23 segregates to these ganglioside-enriched axonal membranes during and after node formation and further that 2) the cell surface expression of ADAM23 is regulated by its interaction with its LGL3/2 ligands, with clearance of ADAM23/Kv1 complexes from the developing nodal and paranodal domains requiring endocytic recycling. It has been proposed before that paranodal environ represents a site of endocytic recycling and sorting of membrane proteins (see Poliak et al., 2001). These mechanisms are currently under investigation in our laboratory.

4. IPI is not defined in the manuscript.

Figure 7J,K and L state that IPI stand for inter-pulse interval. This has now also been remedied in the Figure legend.

Reviewer #2

In this manuscript Kozar-Gillan et al ...demonstrate using genetic mouse models that axonal expression of the metalloproteinase Adam23 is required for initial clustering as well as stability of Kv1 at juxtaparanodes via interaction of Adam23 with LGL2 and LGL3.This is a rigorously conducted, well-written study on a topic of interest to the field. I have only minor comments:

1) *In Figure 1, only colocalization of Kv1.1 with Adam23 was examined. Does Adam23 also colocalize with other Kv1 channels?*

The other Kv1 channel present in normal axons is Kv1.2 which is incorporated with Kv1.1 into hetero-tetrameric voltage gated potassium channels (Wang et al., 1993). Not surprisingly, ADAM23 also co-localises with Kv1.2 in the juxtaparanodal domain. This co-localisation of ADAM23 and Kv1.2 is evident in figure 3B. Expression of Kv1.6 and Kv1.4 at the juxtaparanodal domain has been described in neuromas (Calvo et al., 2016). We did not detect Kv1.4 or Kv1.6 in wildtype or ADAM23 knock out axons with the antibodies available to us (from Neuromab, the same antibodies used in the Calvo et al study).

2) *It would be helpful if the authors provide an expanded discussion of how their findings relate to previous studies of Kv1 knockout mice.*

This point was also raised by reviewer #1 (see above).

3) *The CASPR2 immunostaining in Figure 3A looks like it might be reduced in the Adam23 ScKO. Is that the case?*

The staining was not particularly good for this antibody. Unfortunately, we don't have alternative images available. We don't think that CASPR2 levels are significantly affected by the Schwann cell specific deletion of ADAM23. We further note that all other

components of the juxtaparanodal Kv1 complex examined here (ADAM23, Kv1.1, Kv1.2 and LGL3) are not affected by the Schwann cell specific deletion of ADAM23.

4) Figure 3B does not show staining in L4-5 sensory roots from control animals, only the (nice) internal control of non Cre-expressing Adelta fibers. It would be best to also be able to compare the PV+ axons from cKO and control animals.

We chose to present the sensory root images of the conditional knock out *Adam23^{PvKO/PvKO}* as they contain an internal control. Internal controls are preferable over controls that are subjected to an identical but separate experimental procedure. This controls for fluctuations in fixation, teasing of the nerves and staining with antibodies. The aim of the experiment was to ask whether Kv1 channels would accumulate over time in the absence of ADAM23. The strategy we chose was to selectively delete ADAM23 in a subpopulation of sensory axons (proprioceptive) using the Parv-Cre driver line, which allowed us to exactly address that question. We did not stain the axons with a Parvalbumin antibody as it has been demonstrated before that these proprioceptive neurons are Parvalbumin positive (de Nooij et al., 2013; Hippenmeyer et al., 2005)

5) Could the authors expand on the method by which PV+ axons were identified in Figure 3B? This seems important as there appear to be about 10% of axons without or with ambiguous Adam23 immunostaining JXPs at 16 weeks of age (Figure 4B), so the lack of Adam23 may not be sufficient to identify the axon as PV+. The authors state axon diameter is used to identify PV+ axons, but the exact criteria is not given. Even better, was costaining done to confirm these axons are in fact PV+ axons?

We did not co-stain the axons in Figure 3B with Parvalbumin (see also our comments to point 4). We examined ADAM23 negative versus ADAM23 positive axons and noted that in the absence of ADAM23 no Kv1 channel complexes accumulate at the JXP, whereas ALL nodes positive for either Kv1.1, Kv1.2, CASPR2 or LGL3 were positive for ADAM23. This is also true for the data presented in figure 4C. The fraction of axons that were scored as negative or ambiguous for ADAM23 were also negative or ambiguous for Kv1.1 (and other components tested). We realise that the way we presented those data in bar diagrams is potentially confusing. A line has been added in the legend to figure 4 to make the point that in no instance did we found Kv1.1/Kv1.2/CASPR2 or LGL3 positive JXPs that were negative for ADAM23.

The ADAM23 negative and ambiguous nodes are found in the lower range of axonal diameters and is reflective of the scaling of Kv1 channel complex immune-fluorescence intensity with axonal diameter. Larger diameter myelinated axons have readily detectable levels of Kv1 complexes at the JXP whereas lower calibre axons have less intense JXP staining. Thus, detection of Kv1 complexes in these lower calibre axons is more sensitive to antibody affinity, fixation sensitivity (of the antibody) and artifacts caused by mechanical teasing of the nerve fibers. As such, some of these small calibre axons slip below the

detection level of the JXP antibodies used and they are subsequently scored as negative or ambiguous.

6) *The CASPR staining appears more diffuse in the Adam23 PvKO PV+ neurons in Figure 3B. Is that the case?*

Yes, this diffuse staining seems especially evident in the very large calibre axons. However, not all Adam23 negative nodes show this diffuse staining suggesting that it might be a feature of a subset of proprioceptive neurons. In future, it would be of interest to establish whether this nodal morphology segregates with subset (Ia/II or Ib) of proprioceptive neurons.

7) *Some of the text references for Figure 2 appear to be incorrect.*

Thank you, these have now been corrected

Reviewer #3

The present manuscript focuses on the mechanisms underlying Kv1 channel clustering..... This is an interesting study, in particular as there was until now a lack of clear data on juxtaparanodal Kv channel role in axonal conduction properties. The manuscript is very well written and illustrated. I have only few comments and suggestions regarding this work.

1. *Kv channel accumulation at paranodes is slightly delayed compared to paranodal and nodal clustering during development. It would have been interesting to show whether ADAM23 follows a similar pattern, by looking at slightly younger ages (P9 and P6?). Also, could Kv1 briefly accumulate in the nodal area prior to be retrieved or to diffuse in ADAM23 KO?*

Previously we examined P8 nerves and found that ADAM23 and Kv1 channels always overlap in their expression (Annelies van den Bogaard and Dies Meijer unpublished). At this stage of postnatal development Kv1/ADAM23 complexes overlap with the paranode in a significant portion of nodes ($\pm 50\%$). However, our analysis is still not complete, and we agree with this reviewer that earlier timepoints need to be examined. We are actively investigating this question

2. *Some of the WT images (ADAM23, Figure 1C and Kv1.2, Figure 1D) are suboptimal.*

We don't think that images are suboptimal in that they show partial overlap between Kv1/ADAM23 complexes with the paranodal domain in some nodes (see point above). It reflects the ongoing segregation of paranodal and juxtaparanodal components which is only fully established in all myelinated axons by the third week of postnatal development (even later in the rat; see Vabnick et al., 1999). With respect to the previous point of this reviewer it is of interest that no Kv1 channel complex components are detected in ADAM23 KO nerves

3. Line 189: why look at P12 and not P10 as before? This question applies for other experiments implicating one of these two timepoints along the study. This should be clarified in the text.

The reason why we looked at P10 for the Adam23KO nerves and P12 for Lg12, Lg13 and Lg12/Lg13 double knock outs is unfortunately rather prosaic. We initiated this analysis aiming to describe the Kv1 channel complexes in nerves of the different genotypes at P12. However, we had to redefine the humane endpoint for ADAM23 pups given the severity of the phenotype. We therefore collected nerves in these animals at P10. This is now explained in the text.

4. Line 200: the mislocalization of juxtaparanodal markers in the paranode could suggest a paranodal alteration. This should maybe be discussed. Also, with aging, juxtaparanodal components tend to relocate to paranodes. Could LG13/ADAM23 be implicated in this process?

We have discussed the potential subtle alterations of the paranode in Adam23 Δ 1/ Δ 1 nerves in response to point 2 reviewer 1.

The age-related partial relocation of Kv1 channel complexes to the paranodal domain is accompanied with ultrastructural alterations of the paranodal axon-glia junctions (Hinman et al., 2006). This might represent a partial phenocopy of Caspr or Contactin knock outs in which the axon-glia boundary function is compromised and Kv1 channel complexes occupy paranodal territory. Whether this involves an active mechanism (as opposed to redistribution into lipid raft domain otherwise occupied by Caspr/Contactin) is an interesting suggestion. Whether this is the case can be investigated by the neuron-specific deletion of Lg13 in young adult animals and following Kv1 complex distribution of time. This is an interesting future line of inquiry.

5. Page 14 : the notion of axonal targeting may confuse the readers. The authors should maybe clarify that it does not refer to axonal membrane targeting.

We have now clarified that by axonal targeting we mean the selective distribution of Kv1 complexes into the axonal domain (as opposed to the soma or dendritic domain) not the axonal membrane.

6. Regarding the role of JXP Kv1 complex in the physiology of the axon, can a paranodal defect be excluded in ADAM23PvKO/PvKO animals, as it could participate in electrophysiological alterations?

We refer to our response to point2 or reviewer 1. We argue that paranodal alterations in Adam23 and Lg13 knock out animals are not evident as nerve conduction velocities are not affected in these animals. Given the complexities of nodal structure and the reported wildly different effects of 4-AP on AP shape in different nerves, our conclusions are presented with caution as we realise that we have only examined a subset of primary sensory axons.

Minor comments:

Could the authors discuss the lack of compensation between ADAM11 and ADAM23?

Although Adam11 is highly expressed in DRG neurons and motor neurons in the spinal cord, we could not detect Adam11 at the JXP with the antibodies available to us (Neuromab N441/35, in house rabbit ADAM11 antibody and a commercial rabbit ADAM11 antibody). These antibodies readily detect ADAM11 at the cerebellar Pinceau.

It is of interest to note that Kole and colleagues describe normal juxtaparanodal Kv1.1/Kv1.2 distribution in peripheral axons of Adam11 knock out animals. This is in line with our finding that LGI2 (which binds strongly to ADAM11) expression at the JXP in spinal cord axons is unaffected by Adam11 deletion (see figure).

It is unclear why ADAM11 cannot compensate for lack of ADAM23 (should ADAM11 indeed be expressed at the JXP) but this is a consistent theme among ADAM11/22/23 and LGI1/2/3/4 interactions. For example, we have shown that ADAM23 cannot compensate for ADAM22 in regulating myelination of peripheral nerves nor can LGI1/2/3 compensate for LGI4 as a ADAM22 ligand in that same process (Özkaynak et al., 2010; Kegel et al., 2014). For LGI4 we have demonstrated that the defining difference between LGI1 and LGI4 lies in a small motif of amino-acids unique to LGI4 and which when conferred to LGI1 bestows a myelination stimulating activity on the mutant protein. Thus, even though LGI ligands all bind to the ADAM11/22/23 receptors the biological outcome of these interactions is unique to the biological process under consideration. So, the function of ADAM23 uniquely depends on its interaction with LGI2/3 and ADAM22 nor ADAM11 can substitute ADAM23. It is likely that the individual combinations of receptor and ligand create a unique interface that is tailored to the specific function of the receptor/ligand pair. To determine what these unique interaction interfaces are and with what components they interact is an essential step towards a complete mechanistic understanding of the fascinating functions these receptor/ligand pairs are involved in and presents a major challenge for the future.

There is a typo line 213.

Thank you for spotting this. This has now been corrected.

December 23, 2022

RE: JCB Manuscript #202211031R

Prof. Dies N Meijer
University of Edinburgh
Centre for Discovery Brain Sciences
49 Little France Crescent
Edinburgh EH16 4SB
United Kingdom

Dear Prof. Meijer:

Thank you for submitting your revised manuscript entitled "LGI3/2-ADAM23 interactions cluster Kv1 channels in myelinated axons to fine-tune axonal properties". Two of the original reviewers have now assessed your revised manuscript and, as you can see, they are satisfied with revisions. Thus, we would be happy to publish your paper in JCB pending final revisions necessary to meet our formatting guidelines (see details below).

To avoid unnecessary delays in the acceptance and publication of your paper, please read the following information carefully. Please go through all the formatting points paying special attention to those marked with asterisks.

A. MANUSCRIPT ORGANIZATION AND FORMATTING:

1) Text limits: Character count for Articles and Tools is < 40,000, not including spaces. Count includes title page, abstract, introduction, results, discussion, and acknowledgments. Count does not include materials and methods, figure legends, references, tables, or supplemental legends.

2) Figures limits: Articles and Tools may have up to 10 main text figures.

Please note that main text figures should be provided as individual, editable files.

3) Figure formatting:

Molecular weight or nucleic acid size markers must be included on all gel electrophoresis.

***** Scale bars must be present on all microscopy images, including inset magnifications. Please add scale bars to Figs 1B-C and 2B.**

Also, please avoid pairing red and green for images and graphs to ensure legibility for color-blind readers. If red and green are paired for images, please ensure that the particular red and green hues used in micrographs or graphs are distinctive with any of the colorblind types. If not, please modify colors accordingly or provide separate images of the individual channels.

4) Statistical analysis:

***** Error bars on graphic representations of numerical data must be clearly described in the figure legend. Please describe errors bars in Fig 5B.**

***** The number of independent data points (n) represented in a graph must be indicated in the legend -please add 'n' for Fig 5B. Please, indicate whether 'n' in Fig 8C-E, 8H-I refers to technical or biological replicates (i.e. number of analyzed cells, samples or animals, number of independent experiments).**

If independent experiments with multiple biological replicates have been performed, we recommend using distribution-reproducibility SuperPlots (please, see Lord et al., JCB 2020) to better display the distribution of the entire dataset, and report statistics (such as means, error bars, and P values) that address the reproducibility of the findings.

Statistical methods should be explained in full in the materials and methods in a separate section.

For figures presenting pooled data the statistical measure should be defined in the figure legends.

Please also be sure to indicate the statistical tests used in each of your experiments (both in the figure legend itself and in a separate methods section) as well as the parameters of the test (for example, if you ran a t-test, please indicate if it was one- or two-sided, etc.).

*** As you used parametric tests in your study (i.e. t-tests), you should have first determined whether the data was normally distributed before selecting that test. In the stats section of the methods, please indicate how you tested for normality. If you did not test for normality, you must state something to the effect that "Data distribution was assumed to be normal but this was not formally tested."

5) Abstract and title:

The abstract should be no longer than 160 words and should communicate the significance of the paper for a general audience.

*** The title should be less than 100 characters including spaces. Make the title concise but accessible to a general readership. To convey the advance more clearly we would like to suggest the following title: "LGI3/2-ADAM23 interactions cluster Kv1 channels in myelinated axons to regulate their refractory periods."

6) Materials and methods:

Should be comprehensive and not simply reference a previous publication for details on how an experiment was performed. The text should not refer to methods "...as previously described."

Also, the materials and methods should be included with the main manuscript text and not in the supplementary materials.

7) For all cell lines, vectors, constructs/cDNAs, etc. - all genetic material: please include database / vendor ID (e.g., Addgene, ATCC, etc.) or if unavailable, please briefly describe their basic genetic features, even if described in other published work or gifted to you by other investigators (and provide references where appropriate).

Please be sure to provide the sequences for all of your oligos: primers, si/shRNA, RNAi, gRNAs, etc. in the materials and methods.

*** You must also indicate in the methods the source, species, and catalog numbers/vendor identifiers (where appropriate) for all of your antibodies, including secondary. If antibodies are not commercial, please add a reference citation if possible.

8) Microscope image acquisition:

The following information must be provided about the acquisition and processing of images:

a. Make and model of microscope

b. Type, magnification, and numerical aperture of the objective lenses

c. Temperature

d. imaging medium

e. Fluorochromes

*** f. Camera make and model

g. Acquisition software

h. Any software used for image processing subsequent to data acquisition. Please include details and types of operations involved (e.g., type of deconvolution, 3D reconstitutions, surface or volume rendering, gamma adjustments, etc.).

10) Supplemental materials:

There are strict limits on the allowable amount of supplemental data. Articles/Tools may have up to 5 supplemental figures. There is no limit for supplemental tables.

*** Please note that supplemental figures and tables should be provided as individual, editable files.

*** Supplemental Figure legends seem missing.

A summary of all supplemental material should appear at the end of the Materials and Methods section (please see any recent JCB paper for an example of this summary).

11) Video legends: Should describe what is being shown, the cell type or tissue being viewed (including relevant cell treatments, concentration and duration, or transfection), the imaging method (e.g., time-lapse epifluorescence microscopy), what each color represents, how often frames were collected, the frames/second display rate, and the number of any figure that has related video stills or images.

12) eTOC summary:

A ~40-50 word summary that describes the context and significance of the findings for a general readership should be included on the title page.

*** The statement should be written in the present tense and refer to the work in the third person. It should begin with "First author name(s) et al..." to match our preferred style.

13) Conflict of interest statement:

JCB requires inclusion of a statement in the acknowledgements regarding competing financial interests. If no competing financial interests exist, please include the following statement: "The authors declare no competing financial interests."

14) Author contribution section:

A separate author contribution section is required following the Acknowledgments in all research manuscripts.

*** All authors should be mentioned and designated by their first and middle initials and full surnames and the CRediT nomenclature is encouraged (<https://casrai.org/credit/>).

15) ORCID IDs: ORCID IDs are unique identifiers allowing researchers to create a record of their various scholarly contributions in a single place. At resubmission of your final files, please consider providing an ORCID ID for as many contributing authors as possible.

16) Materials and data sharing:

All animal and human studies must be conducted in compliance with relevant local guidelines, such as the US Department of Health and Human Services Guide for the Care and Use of Laboratory Animals or MRC guidelines, and must be approved by the authors' Institutional Review Board(s). A statement to this effect with the name of the approving IRB(s) must be included in the Materials and Methods section.

*** As a condition of publication, authors must make protocols and unique materials (including, but not limited to, cloned DNAs; antibodies; bacterial, animal, or plant cells; and viruses) described in our published articles freely available upon request by researchers, who may use them in their own laboratory only. All materials must be made available on request and without undue delay. We strongly encourage to deposit all the cell lines/strains and reagents generated in this study in public repositories.

All datasets included in the manuscript must be available from the date of online publication, and the source code for all custom computational methods, apart from commercial software programs, must be made available either in a publicly available database or as supplemental materials hosted on the journal website. Numerous resources exist for data storage and sharing (see Data Deposition: <https://rupress.org/jcb/pages/data-deposition>), and you should choose the most appropriate venue based on your data type and/or community standard. If no appropriate specific database exists, please deposit your data to an appropriate publicly available database.

17) Please note that JCB now requires authors to submit Source Data used to generate figures containing gels and Western blots with all revised manuscripts. This Source Data consists of fully uncropped and unprocessed images for each gel/blot displayed in the main and supplemental figures. The Source Data files will be directly linked to specific figures in the published article.

Since your paper includes cropped gel and/or blot images, please be sure to provide one Source Data file for each figure that contains gels and/or blots along with your revised manuscript files. File names for Source Data figures should be alphanumeric without any spaces or special characters (i.e., SourceDataF#, where F# refers to the associated main figure number or SourceDataFS# for those associated with Supplementary figures). The lanes of the gels/blots should be labeled as they are in

the associated figure, the place where cropping was applied should be marked (with a box), and molecular weight/size standards should be labeled wherever possible.

B. FINAL FILES:

Thank you for this interesting contribution, we look forward to publishing your paper in Journal of Cell Biology.

Sincerely,

Marc Freeman
Monitoring Editor
Journal of Cell Biology

Lucia Morgado-Palacin, PhD
Scientific Editor
Journal of Cell Biology

Reviewer #2 (Comments to the Authors (Required)):

The authors have addressed my concerns.

Reviewer #3 (Comments to the Authors (Required)):

The present work shows that ADAM23 is required for both accumulation and maintenance of Kv channel complexes at the juxtaparanodes, and that this depends on ADAM23 interaction with its ligands LGI3/2. It further shows that juxtaparanodal Kv channels affect the refractory period, suggesting that they could participate in enabling the high frequency firing of action potentials. This work is of high quality and interest, the data are very convincing and the paper is well written. The authors have addressed my (minor) concerns in the revised manuscript and I think it is adequate in its present form for publication in JCB.